# Serum apolipoprotein A-I potentiates the therapeutic efficacy of lysocin E against *Staphylococcus aureus*

Hiroshi Hamamoto [1,2,10], Suresh Panthee [3,10], Atmika Paudel [4], Kenichi Ishii[5], Jyunichiro Yasukawa[6], Jie Su[7], Atsushi Miyashita[1], Hiroaki Itoh [8], Kotaro Tokumoto[8], Masayuki Inoue [8] & Kazuhisa Sekimizu [3,9] ✉

Lysocin E is a lipopeptide with antibiotic activity against methicillin-resistant *Staphylococcus aureus*. For unclear reasons, the antibacterial activity of lysocin E in a mouse systemic infection model is higher than expected from in vitro results, and the in vitro activity is enhanced by addition of bovine serum. Here, we confirm that serum from various species, including humans, increases lysocin E antimicrobial activity, and identify apolipoprotein A-I (ApoA-I) as an enhancing factor. ApoA-I increases the antibacterial activity of lysocin E when added in vitro, and the antibiotic displays reduced activity in *ApoA-I* gene knockout mice. Binding of ApoA-I to lysocin E is enhanced by lipid II, a cell-wall synthesis precursor found in the bacterial membrane. Thus, the antimicrobial activity of lysocin E is potentiated through interactions with host serum proteins and microbial components.

[1] Teikyo University Institute of Medical Mycology, Tokyo, Japan. [2] Division of Sport and Health Science, Graduate School of Medical Care and Technology, Teikyo University, Tokyo, Japan. [3] Drug Discoveries by Silkworm Models, Faculty of Pharma-Science, Teikyo University, Tokyo, Japan. [4] International Institute for Zoonosis Control, Hokkaido University, Sapporo, Japan. [5] Department of Biological Sciences, Graduate School of Science, The University of Tokyo, Tokyo, Japan. [6] Department of Biochemistry, Faculty of Pharmaceutical Sciences, Doshisha Women's College of Liberal Arts, Kyoto, Japan. [7] National Marine Environmental Monitoring Center, Dalian, China. [8] Graduate School of Pharmaceutical Sciences, The University of Tokyo, Tokyo, Japan. [9] Genome Pharmaceuticals Institute, Ltd, Tokyo, Japan. [10] These authors contributed equally: Hiroshi Hamamoto, Suresh Panthee. ✉email: sekimizu@main.teikyo-u.ac.jp

Since the discovery of penicillin, antimicrobial agents have been screened on the basis of their in vitro antimicrobial activity, which does not accurately reflect the host environment. Protein binding affects the activity of antimicrobials. For example, some host factors, such as serum albumin, bind to some antimicrobials, which influences their activity. Indeed, the activity of many antimicrobials is decreased by adding blood serum to the culture medium due to the decreased free-drug concentration resulting from large distribution[1]. On the other hand, the activities of some antimicrobials, such as azithromycin, are enhanced by serum factors in the blood[2]. Daptomycin and telavancin exhibit high protein binding capacities, although these high protein binding capacities do not affect their therapeutic efficacy[3,4]. Evaluation of the therapeutic efficacy of antimicrobials in vivo is therefore crucial.

Our unique strategy to search for antimicrobial agents, utilising silkworms as a host for evaluating the in vivo effects of antimicrobials at early stages of screening[5,6], allows for the exclusion of compounds with reduced antimicrobial activity due to host protein binding[7,8], and led to the discovery of a novel antibiotic, lysocin E[9] (Supplementary Fig. 1a). Lysocin E targets menaquinone, a cofactor in the electron transport chain, in the cell membrane of Staphylococcus aureus and disrupts the membrane, which results in rapid bactericidal activity. The total synthesis of lysocin E has been achieved[10] and comprehensive amino acid substitution by the "one-bead-one-compound" method revealed that the membrane disruption of liposomes containing menaquinone is well correlated with the antimicrobial activity of lysocin E derivatives against S. aureus[11]. In a previous study, we demonstrated that lysocin E is therapeutically effective against the methicillin-sensitive S. aureus (MSSA) Smith strain at lower doses than vancomycin, despite its lower antibacterial activity compared with vancomycin[9]. In addition, we found that the antimicrobial activity of lysocin E in vitro is highly increased by adding bovine serum to the medium[11]. We, therefore, hypothesised that the antimicrobial activity of lysocin E is promoted by factors in the host environment.

Here, we report the identification of ApoA-I and ApoA-II as enhancing factors in serum for lysocin E antimicrobial and therapeutic activity. We further demonstrate that ApoA-I promotes membrane-damaging and bactericidal activity of lysocin E at a sub-minimum inhibitory concentration (MIC) against S. aureus and lipid II mediates the binding of lysocin E to ApoA-I. Furthermore, the antimicrobial activity of nisin, which is also lipid II-binding antimicrobial, is enhanced by ApoA-I and lipid II interaction. Overall, our study demonstrates the interaction of host factor(s) and the bacterial component can potentiate these antimicrobial activities.

## Results

**Serum ApoA-I enhances lysocin E activity**. We found that the ratio of the median effective dose to the MIC ($ED_{50}$/MIC) is more than 16-fold smaller for lysocin E compared with the other anti-MRSA drugs, indicating that lysocin E has a high therapeutic efficacy compared with the other anti-MRSA drugs (Table 1). In contrast to clinically used anti-MRSA microbials such as

vancomycin and linezolid, lysocin E (**1**) has high protein binding capacity (Table 1). The antimicrobial activity of lysocin E against MSSA and MRSA strains, however, could be increased by adding sera from various species, including human or silkworm hemolymph (Fig. 1a, Table 1 and Supplementary Table 1). The increased antimicrobial activity of lysocin E induced by bovine serum was also observed against other Gram-positive bacteria such as Staphylococcus spp. and Bacillus spp. (Supplementary Table 1). To understand the mechanism by which serum enhances the bactericidal activity of lysocin E, we purified the responsible factors in bovine serum by measuring the increase in antimicrobial activity of lysocin E due to exposure to each of the various serum components with activity guided fractionation (Supplementary Fig. 2a). Ethanol extraction, octadecyl silica (ODS) column chromatography and gel filtration chromatography were performed. Analysis by sodium dodecyl sulfate-polyacrylamide gel electrophoresis (SDS-PAGE) revealed that the elution patterns of 32 and 10 kDa proteins from gel filtration column chromatography were consistent with the antimicrobial-enhancing activity of lysocin E (Supplementary Fig. 2b, c). Peptide mass fingerprinting analysis revealed that the proteins were bovine ApoA-I (bApoA-I) and ApoA-II (Supplementary Fig. 2d). Adding recombinant human ApoA-I (rhApoA-I) or purified human ApoA-II (hApoA-II) to the culture medium reduced the MIC of lysocin E (Fig. 1b). In addition, the specific activity of rhApoA-I and recombinant mouse ApoA-I (rmApoA-I) was similar to that of bApoA-I (Supplementary Fig. 2e), confirming that ApoA-I and ApoA-II are serum factors responsible for enhancing the antimicrobial activity of lysocin E. rhApoA-I decreased the MIC against various MSSA and MRSA strains, including the highly virulent USA300 JE2 strain (Supplementary Table 2). Although ApoA-I exhibits antimicrobial activity against Gram-negative bacteria such as Escherichia coli and Klebsiella pneumoniae[12,13], the addition of up to 300 or 600 μg ml$^{-1}$ of purified bovine and human ApoA-I or ApoA-II did not inhibit S. aureus growth (Supplementary Fig. 2f), as previously reported[13]. Thus, the enhanced antimicrobial activity of lysocin E induced by ApoA-I and ApoA-II is not due to the direct antimicrobial activity of ApoA-I and ApoA-II against S. aureus.

To obtain genetic evidence for the involvement of ApoA in the therapeutic effects of lysocin E, we examined the therapeutic effect of lysocin E using gene knockout mice[14]. In mouse plasma, the ApoA-I concentration is ~7-fold higher than that of ApoA-II[15]. In addition, bovine and human ApoA-I and ApoA-II similarly enhanced the antimicrobial activity of lysocin E (Supplementary Fig. 2a, e). Further, we confirmed that mixing rhApoA-I and hApoA-II in a similar concentration ratio in human blood had no synergistic effects (Supplementary Fig. 2g), suggesting that these factors have similar antimicrobial-enhancing effects and work in an independent manner. Therefore, we performed further analysis using Apoa1 gene knockout mice. The enhanced antimicrobial activity of lysocin E in the serum of Apoa1 homozygous-deficient mice was 6-fold lower than that in wild-type mice (Fig. 1c). The remaining activity of the serum in the Apoa1 gene-deficient mice could be explained by ApoA-II activity

**Table 1 Therapeutic effect in a mouse systemic MRSA infection model and antimicrobial activity of anti-MRSA agent against MRSA.**

|  | Lysocin E | Vancomycin | Linezolid | Daptomycin |
|---|---|---|---|---|
| Antimicrobial activity (MIC: μg ml$^{-1}$) | 4 | 1 | 2 | 0.5 |
| Therapeutic effect ($ED_{50}$: mg kg$^{-1}$) | 0.36 | 4.2 | 3.8 | 0.73 |
| $ED_{50}$/MIC | 0.098 | 4.2 | 1.9 | 1.6 |
| Protein binding (%, human plasma) | >99 | 10–50[51] | 18[52] | 90–96[52] |
| MIC in 10% bovine serum | 0.063 | 1 | 2 | 1 |

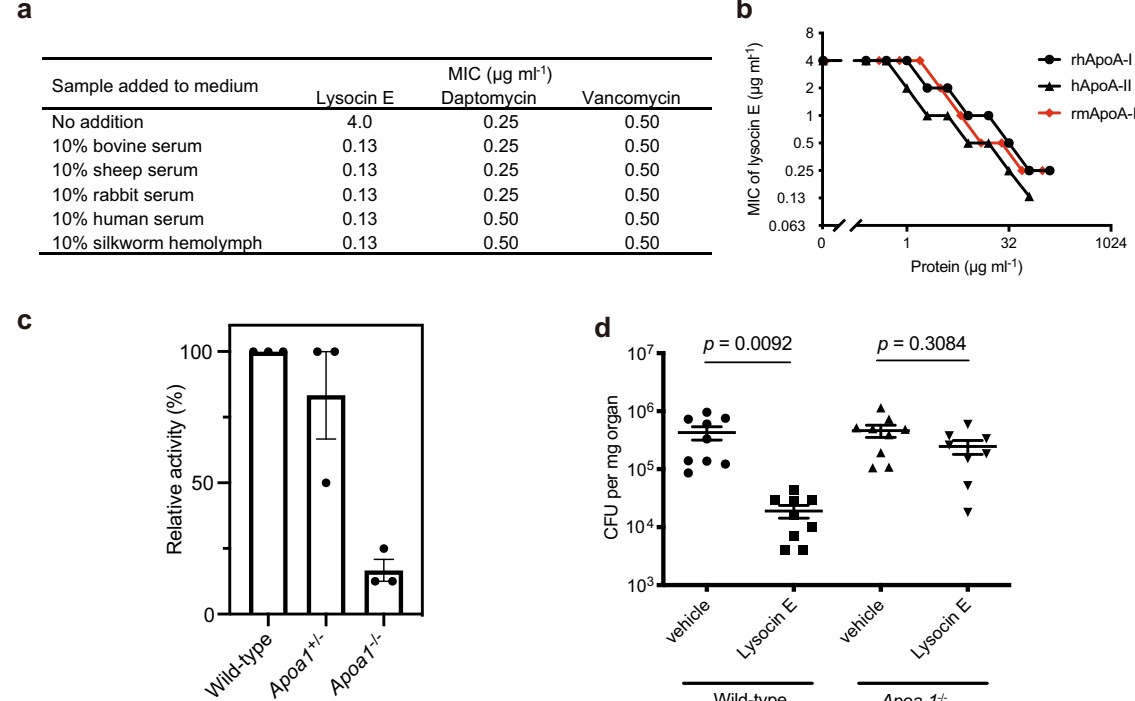

**Fig. 1 Apolipoprotein A enhances the antimicrobial and therapeutic activity of lysocin E. a** Effect of serum from various animals on the antimicrobial activities of lysocin E, vancomycin and daptomycin against *S. aureus* MSSA1 strain. The same results were obtained in three independent experiments. **b** Reduction of the MIC of lysocin E against *S. aureus* by the addition of rhApoA-I (black circle), hApoA-II (black triangle) and rmApoA-I protein (red diamond). Data are representative of duplicate experiments with similar results. **c** The lysocin E-enhancing activity of serum prepared from *Apoa1*-deficient mice. Data represent the mean ± SEM from three independent experiments. **d** Effect of lysocin E in a systemic *S. aureus* infection model using *Apoa1*-deficient mice. Wild-type and *Apoa1* homozygous-deficient mice were infected with the Newman strain and the number of viable bacteria in the kidney 1 day after infection was compared between the lysocin E- and vehicle-administered groups (n = 9/group). Data represent the mean ± SEM and statistical analysis was performed using a one-way ANOVA with Tukey's multiple comparison test. The data are representative of duplicate experiments with similar results.

because ApoA-II is present in the serum of *Apoa1* homozygous-deficient mice at the same level as in wild-type mice[16]. Next, we examined the therapeutic efficacy of lysocin E in an *S. aureus* systemic infection model using *Apoa1* homozygous-deficient mice. Treatment with lysocin E did not reduce the number of surviving bacteria in the kidney of *Apoa1* homozygous-deficient mice, whereas it did reduce the number of bacteria in the kidney of wild-type mice (Fig. 1d). These results suggest that ApoA-I contributes to the therapeutic effect of lysocin E in mice.

**ApoA-I activity is mediated via lipid II.** ApoA-I and high-density lipoprotein (HDL) containing ApoA-I bind to the surface of bacteria[17,18] and free lipoteichoic acid (LTA)[19]. We confirmed that rhApoA-I binds to the surface of *S. aureus* in a concentration-dependent manner (Fig. 2a and Supplementary Fig. 3a). Because adding rhApoA-I did not increase the amount of lysocin E bound to the surface of *S. aureus* (Supplementary Fig. 3b), the enhancing effect of ApoA-I cannot be explained by an increased accumulation of lysocin E on the cell surface. To elucidate the lysocin E-enhancing functions of ApoA-I, we searched for lysocin E derivatives that did not respond to the addition of rhApoA-I, but the antimicrobial activity did not differ from that of the natural type in the absence of rhApoA-I. We found that the antimicrobial activity of Bu-type lysocin E (**2**) (Supplementary Fig. 1a)[10], which has a modified fatty acid chain moiety, was not enhanced by rhApoA-I (Supplementary Table 3). We assumed that the interaction of lysocin E Bu-type with rhApoA-I differs from that of the natural-type lysocin

E. We compared the affinities of the natural- and Bu-type lysocin E for rhApoA-I and found that their binding constants were similar, 3.6 and 3.4 μM, respectively (Supplementary Fig. 4a). The results suggested that lysocin E interacts with ApoA-I, although this interaction does not explain the difference in the ApoA-I response between natural- and Bu-type lysocin E. We then tested the binding affinity of the natural- and Bu-type lysocin E to menaquinone, a cofactor of the electron transport chain of *S. aureus* that is a target of lysocin E[9]. The binding constant of natural lysocin E with menaquinone was 4.0 μM, a value similar to the previous report[9], and that of the Bu-type lysocin E was 5.0 μM (Supplementary Fig. 4b). Further, the binding capacity of lysocin E to ApoA-I in a rhApoA-I pull-down assay did not significantly differ between the natural- and Bu-type lysocin E in the presence of menaquinone (Fig. 2b), suggesting that the presence of menaquinone does not affect the affinities of natural- and Bu-type lysocin E for ApoA-I.

Lipid II (**6**, Supplementary Fig. 1c) was recently reported to interact with lysocin E, and treatment of *S. aureus* with sub-MIC of lysocin E leads to the accumulation of lipid II in the *S. aureus* membrane[20]. We hypothesised that lipid II is involved in the lysocin E antimicrobial-enhancing effect of ApoA-I, and tested this hypothesis in the following experiments. We found that natural-type lysocin E had a significantly higher binding capacity to lipid II than Bu-type lysocin E (Supplementary Fig. 4c). We further demonstrated that lipid II increased the binding of the natural-type lysocin E to rhApoA-I (Fig. 2b) in a pull-down assay in the presence of menaquinone, and this phenomenon was not observed in the absence of menaquinone (Fig. 2c). In addition,

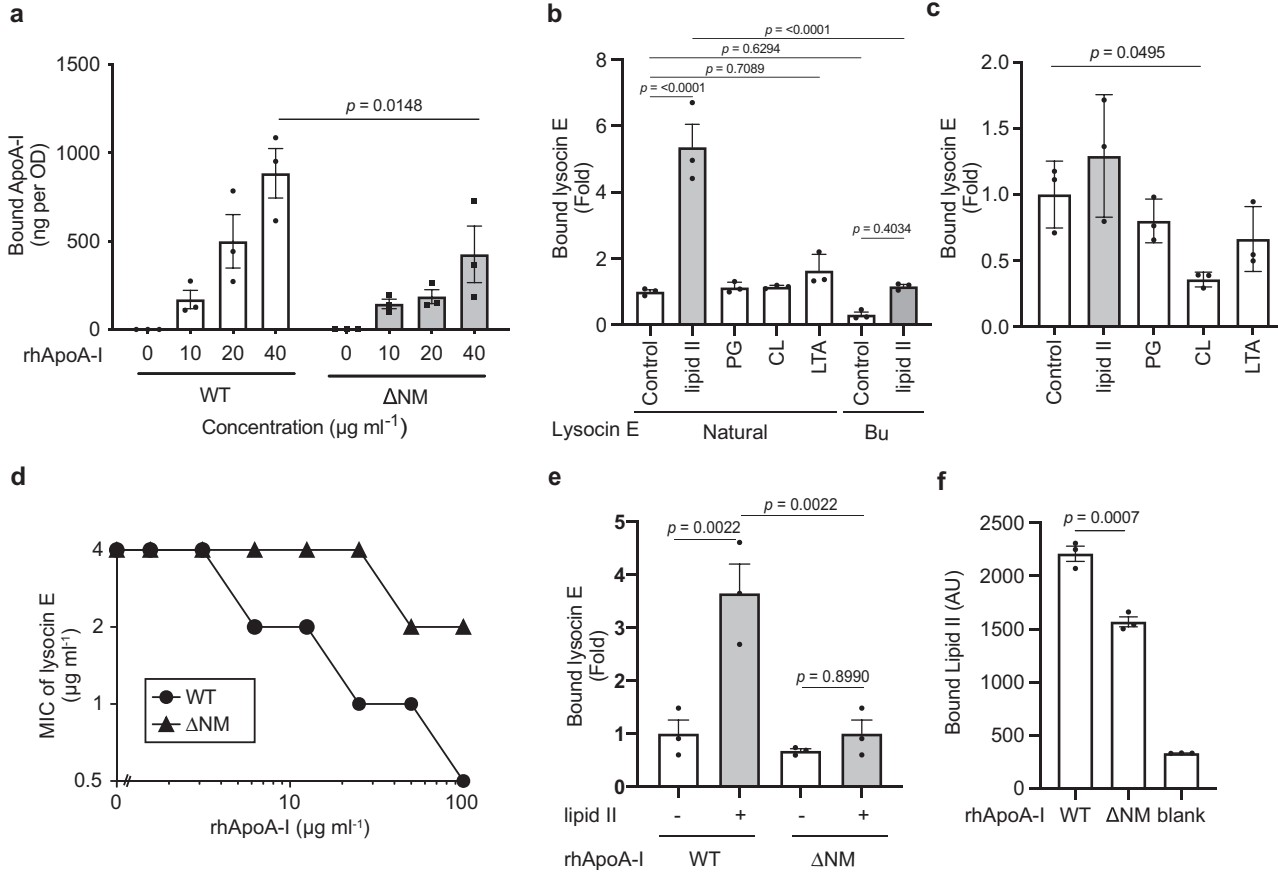

**Fig. 2 Lipid II enhances the interaction of lysocin E with ApoA-I. a** Binding of rhApoA-I wild-type and ΔNM variant to the cell surface of *S. aureus* RN4220. Different concentrations of proteins were added to *S. aureus*. The cells were then treated with SDS-PAGE sample buffer and centrifuged. The supernatant was then boiled and applied to SDS-PAGE and band intensities were quantified using ImageJ. Data represent mean ± SEM from three independent experiments. The statistical analysis was performed by two-way ANOVA with Bonferroni's correction for multiple comparisons. **b** Lipid II-mediated binding of lysocin E to rhApoA-I in a rhApoA-I pull-down assay in the presence of menaquinone. Data represent mean ± SEM of triplicate results. Statistical analysis was performed by one-way ANOVA with Tukey's multiple comparisons test. **c** Lipid II and other ApoA-I binding substances effect on lysocin E binding to rhApoA-I in the absence of menaquinone. Data represent mean ± SEM from three independent experiments and statistical analysis was performed by one-way ANOVA with a post hoc Dunnett multiple comparisons test against the control. **d** rhApoA-I ΔNM decreased the lysocin E antimicrobial activity-enhancing effect against *S. aureus* RN4220 compared with wild-type rhApoA-I. MIC values were determined by the microdilution assay described in the "Methods" and representative data are shown from triplicate independent experiments with similar results. **e** rhApoA-I ΔNM reduced lipid II-mediated lysocin E binding to rhApoA-I in the presence of menaquinone. Data represent mean ± SEM of triplicate results. Statistical analysis was performed by one-way ANOVA Tukey's multiple comparison test. **f** Binding capacity of lipid II to rhApoA-I and rhApoA-I ΔNM variant. Data represent mean ± SEM from three independent experiments and statistical analysis was performed by one-way ANOVA Dunnett multiple comparisons test against WT.

other ApoA-I binding lipids, such as phosphatidylglycerol (PG)[21], cardiolipin (CL)[22] and LTA[19], which are present in the *S. aureus* membrane, did not increase natural-type lysocin E binding to rhApoA-I in the presence or absence of menaquinone (Fig. 2b, c). To further confirm the importance of the lipid II–ApoA-I interaction, we constructed rhApoA-I with deletion of amino acids from the N-terminal (1–65 AA; ΔN), middle (146–160 AA; ΔM) and C-terminal (185–243 AA; ΔC) regions, known to be involved in ApoA-I function[23,24] (Supplementary Fig. 5a). All these variants retained at least one-third of their lysocin E-enhancing activity (Supplementary Fig. 5b). Next, we constructed ΔNM and ΔNC variants and found that the ΔNM variant lost the majority of the lysocin E-enhancing activity (Fig. 2d and Supplementary Fig. 5b). We observed little difference in the binding affinity of lysocin E to rhApoA-I ΔNM compared to rhApoA-I wild-type (Supplementary Fig. 4a), while, compared with wild-type rhApoA-I, rhApoA-I ΔNM had lower lipid II-

dependent lysocin E binding capacity (Fig. 2e). These findings suggested that lipid II enhanced the recruitment of the lysocin E complex to ApoA-I.

Nisin is another antimicrobial compound that binds lipid II with membrane-damaging and bactericidal activities against *S. aureus*[25]. We found that the antimicrobial activity of nisin was enhanced by adding rhApoA-I as in the case of lysocin E (Supplementary Table 3). We then demonstrated in a pull-down assay that the binding of nisin to rhApoA-I was increased by lipid II, but not by other ApoA-I binding lipids (Fig. 3a). These findings suggest that the binding of nisin to ApoA-I is mediated by lipid II, and results in enhanced antimicrobial activity. We further tested whether the bactericidal activities of lysocin E and nisin are enhanced by ApoA-I, and found that rhApoA-I enhanced the bactericidal activity of both antimicrobials in sub-MIC conditions (Fig. 3b). We also evaluated whether the membrane-damaging activity and loss of membrane potential

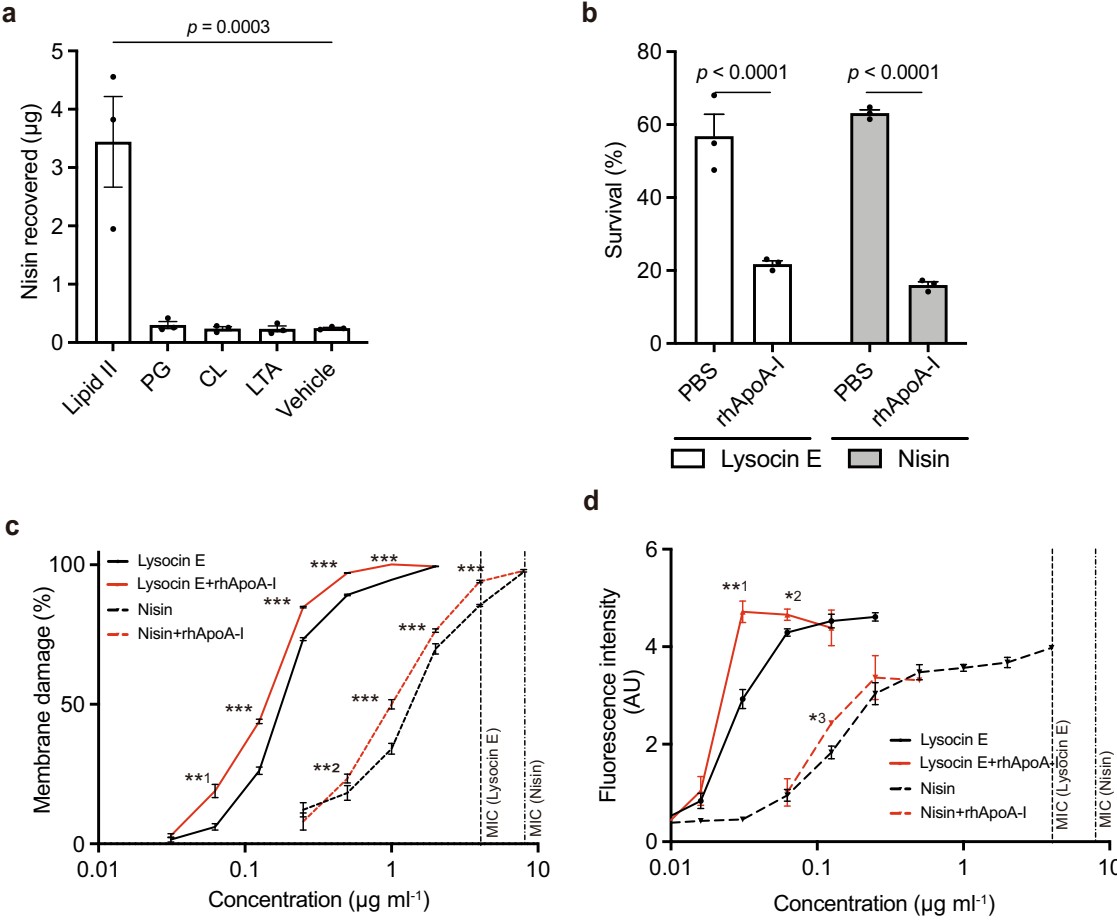

**Fig. 3 Nisin and lysocin E exhibit common characteristics under sub-MIC. a** Increased lipid II-mediated nisin binding to rhApoA-I. Data represent mean ± SEM of triplicate results. Statistical analysis was performed by one-way ANOVA with Tukey's multiple comparisons test. **b** rhApoA-I enhanced the killing ability of lysocin E at sub-MIC. Data represent mean ± SEM of triplicates and statistical analyses were performed using two-way ANOVA Šídák's multiple comparisons test. **c** rhApoA-I enhanced the membrane disruption activity of lysocin E and nisin at sub-MIC. Data represent mean ± SEM of quadruplicate experiments. Statistical analyses were performed using the unpaired Student's two-tailed $t$ test between the values with rhApoA-I and those without rhApoA-I (**[1]$p$ = 0.0025, **[2]$p$ = 0.0144 and ***$p$ < 0.0001). **d** Loss of membrane potential of *S. aureus* after treatment with lysocin E and nisin. Data from triplicate experiments (mean ± SEM) are shown and represent an arbitrary fluorescent unit at 60 s after the addition of the compounds. Statistical analyses were performed using the unpaired Student's two-tailed $t$ test between the values with rhApoA-I and those without rhApoA-I (**[1]$p$ = 0.0004, *[2]$p$ = 0.0213 and *[3]$p$ = 0.0132).

induced by lysocin E and nisin were increased by the addition of ApoA-I. The membrane-damaging activity and loss of membrane potential in *S. aureus* were partially induced by both antimicrobials at sub-MICs and increased by the addition of rhApoA-I (Fig. 3c, d). In addition, externally added lipid II did not affect the antimicrobial activity of lysocin E without rhApoA-I, while it attenuated the lysocin E-enhancing activity of rhApoA-I, suggesting that the interaction of lipid II and ApoA-I is essential for lysocin E function at a sub-MIC (Supplementary Table 3). These findings implied that lysocin E and nisin induce loss of membrane integrity at sub-MIC and that ApoA-I promotes the membrane-disrupting activity of both antimicrobials via lipid II, thereby enhancing the bactericidal actions of both antimicrobials.

**A role of menaquinone in the membrane disruption activity of lysocin E**. We further examined the role of menaquinone in the presence of lipid II in the membrane disruptive activity of lysocin E. Santiago et al.[20] reported that lysocin E can be eluted from immobilised lipid II bound to lysocin E by the addition of lipid II, but not by menaquinone. Our finding that lysocin E bound to immobilised menaquinone was eluted by the addition of lipid II,

but not by menaquinone, confirms this observation (Fig. 4a). These results suggested that both lipid II and menaquinone have an affinity for lysocin E, and that lipid II has a higher affinity than menaquinone for lysocin E.

The Δ*menA* and Δ*menB* strains having a menaquinone content <0.1% than that in wild type[9] exhibit resistance against lysocin E. We confirmed that the lipid II content of these mutants was reduced to ~13% compared with wild type (Fig. 4b), but the antimicrobial activity of nisin against these mutants was indistinguishable from that of the wild type (Fig. 4c). In addition, externally added lipid II in the medium did not affect the antimicrobial activity of lysocin E, whereas the antimicrobial activity of nisin was significantly decreased (Supplementary Table 3) by the addition of lipid II. Furthermore, we demonstrated that lipid II in lipid vesicles is not involved in the membrane disruptive activity of lysocin E (Fig. 4d), while nisin disrupts liposomes containing lipid II as previously reported[26], but not liposomes containing menaquinone (Fig. 4e). We assume that lysocin E could not interact with lipid II in the membrane, which is supported by the results that delipidated lipid II (Supplementary Fig. 1d) does not elute lysocin E immobilised on menaquinone (Fig. 4a). On the other hand, menaquinone

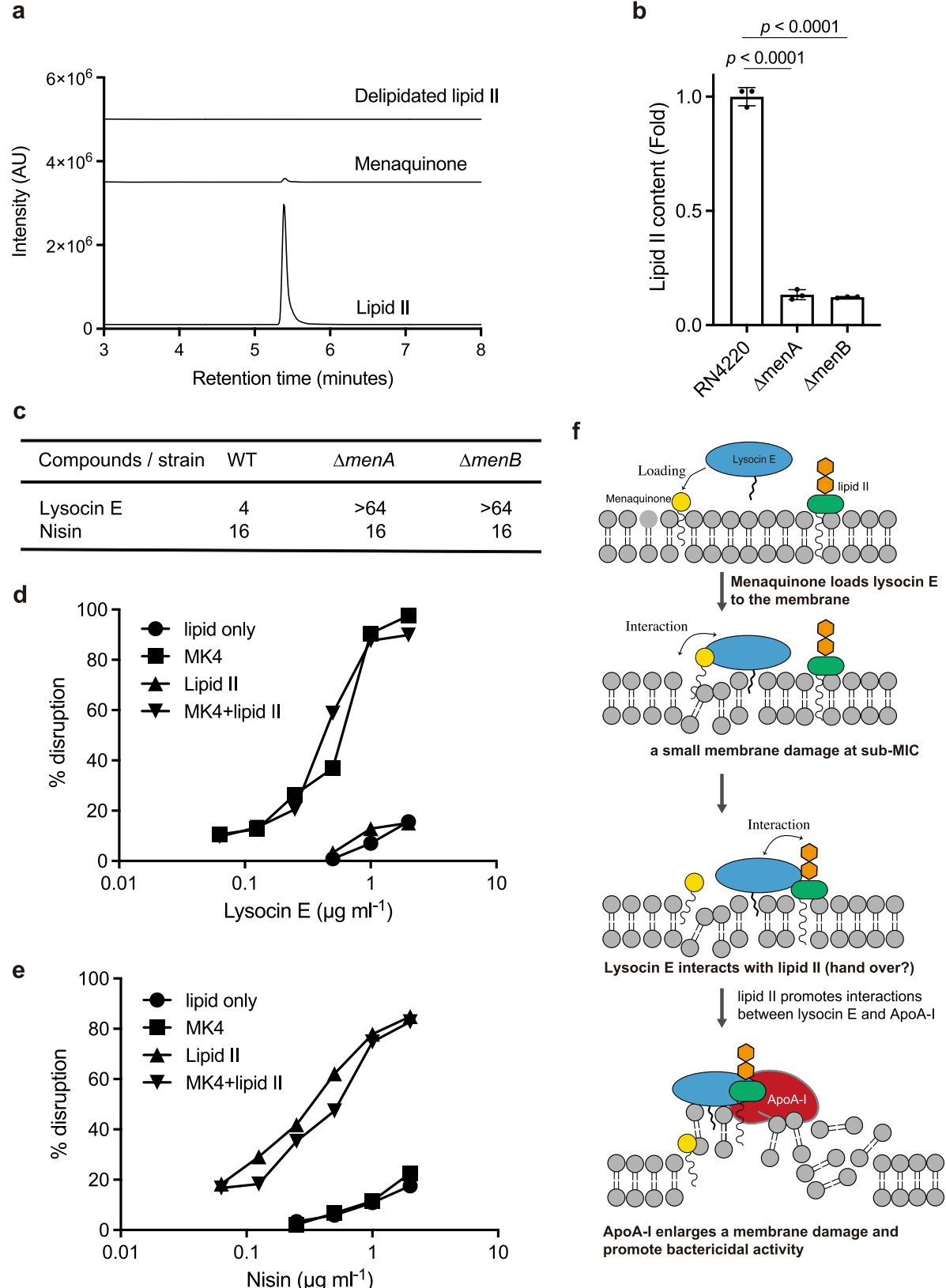

is required for loading lysocin E onto/into the membrane and has essential functions for the membrane disruptive activity of lysocin E. Subsequently, lysocin E might transfer to lipid II and disrupt the membrane further in the presence of ApoA-I (Fig. 4f).

## Discussion

The findings of the present study demonstrated that the anti-microbial activity of lysocin E is enhanced by apolipoproteins in sera, contributing to the therapeutic efficacy of lysocin E (Fig. 1). To our knowledge, this is the first study providing genetic

**Fig. 4 The role of menaquinone and lipid II in membrane-damaging activity. a** Lipid tail of lipid II (**6**) is required to interact with lysocin E (**1**). Streptavidin beads were coated with biotinylated menaquinone 4 (MK-4) (**5**) and allowed to interact with lysocin E (**1**). The beads were then washed twice to remove unbound lysocin E (**1**) and eluted with MK-4, lipid II (**6**) and delipidated lipid II (**7**). The amount of lysocin E (**1**) in the eluate was quantified using UPLC-MS. **b** Lipid II content in the menaquinone biosynthesis disruptant mutants (ΔmenA and ΔmenB) compared with the wild-type RN4220 strain. Data represent mean ± SEM of three independent experiments, and statistical analysis was performed by one-way ANOVA with a post hoc Dunnett multiple comparisons test against wild type. **c** Antimicrobial activity of lysocin E and nisin against ΔmenA and ΔmenB strain. The microdilution assay was performed using a TSB medium. Two independent experiments were performed and the same results were obtained. **d, e** Membrane disruption activity of lysocin E (**d**) or nisin (**e**) on liposomes consisting of lipids, phosphatidylglycerol and cardiolipin, containing menaquinone or/and lipid II. Data are representative of duplicate experiments with similar results. **f** Presumed model of ApoA-I-induced enhancement of the lysocin E antimicrobial activity. First, menaquinone loads lysocin E onto/in the membrane and impairs the membrane. Then, lysocin E is transferred to and interacts with lipid II. Finally, ApoA-I bound on the cell surface of *S. aureus* interacts with the lysocin E–lipid II complex and enhances the membrane damage, thereby killing the bacteria.

evidence that specific host factors enhance the activity of an antimicrobial, thereby contributing to its therapeutic effects. Lysocin E was identified by screening based on its therapeutic efficacy using a silkworm *S. aureus* infection model[9]. Serum and ApoA-I and ApoA-II from various species had enhancing effects on the antimicrobial activity of lysocin E (Fig. 1a and Supplementary Fig. 2a, e). In addition, the antimicrobial activity of lysocin E was also enhanced by silkworm hemolymph (Fig. 1a), suggesting that this phenomenon is conserved widely among animals. Insect hemolymph contains apolipophorin I, II and III, which are involved in the transport of lipids, like mammalian ApoA-I[27,28] and apolipophorin III and human apolipoprotein A-I have similar properties[29]. These insect apolipophorins bind the surface of *S. aureus*[30,31]. We consider that the use of the silkworm infection model to screen for compounds with therapeutic effectiveness led to the discovery of the antibiotic lysocin E, which has unique features.

Here, we present a presumed model in which small amounts of membrane damage induced by lysocin E and nisin trigger a coordinated action between ApoA-I and lipid II, which finally leads to massive membrane disruption, resulting in bactericidal action (Fig. 4f). ApoA-I constitutes a major protein of HDLs in mammals and HDL carries lipids such as cholesterol. In addition, PG and CL, which are acidic phospholipids constituting the cell membrane of *S. aureus*[32], bind ApoA-I[21,22]. Furthermore, a previous study suggested that the C-terminal region of ApoA-I, required for binding lipopolysaccharides, contributes to complement-dependent bactericidal activity against *Yersinia enterocolitica*[23]. From the present study, the addition of sole purified ApoA-I showed enhancing the antimicrobial activity of lysocin E and nisin (Fig. 1b), and PG and CL did not increase binding of lysocin E and nisin to ApoA-I (Figs. 2b, c and 3a), suggesting ApoA-I directly interacted with lipid II and enhanced antimicrobial activity of antimicrobials. Further, C-terminal region had less of an effect on lysocin E antimicrobial activity (Supplementary Fig. 5b), suggesting that functional sites for the lipid II-mediated antimicrobial-enhancing activity of lysocin E differ from this region. In addition, the ApoA-I ΔNM variant had reduced binding capacity against the cell surface of *S. aureus* compared with wild-type ApoA-I (Fig. 2a), suggesting that these regions are involved in the interaction of ApoA-I with the bacterial surface and contribute to the antimicrobial-enhancing activity of lysocin E. Among ten tandem repeat helices present in ApoA-I, H1 (residues 44–65) and H10 (residues 222–243) possess the highest lipid-binding activity[33]. In addition, helix 6 (residues 143–165) is involved in activating lecithin cholesterol acyltransferase and has a low lipid-binding affinity[33]. ΔNM variant, lacking helices 1 and 6, had a reduced lysocin E antimicrobial activity-enhancing effect, whereas the ΔC variant, lacking helix 10, did not exhibit a reduced lysocin E antimicrobial activity-enhancing effect. This finding suggests a complex relationship between the lipid-binding nature of ApoA-I and the induction of

antimicrobial activity mediated by lipid II. On the basis of the recent structural analysis[34], it can be expected that deletion of the N and M regions would severely affect the structure and function of the variant, although the underlying process by which ApoA-I variants form a complex with lipid II and lysocin E, including their structural elucidation, requires further investigation. In this study, we revealed that ApoA-I also interacts with lysocin E (Supplementary Fig. 4a). Even though ApoA-I did not increase lysocin E binding to the cell surface of *S. aureus* (Supplementary Fig. 3b), we cannot deny the possibility that lysocin E forms a complex with ApoA-I before it encounters the microbial surface because ApoA-I is a major lipoprotein of HDL in the blood. ApoA-II is also found in HDL in the blood, and thus these factors may work independently, while how HDL interacts with *S. aureus* and lysocin E and contributes to enhancing the antimicrobial activity of lysocin E were not elucidated in this study. Furthermore, the equilibrium dissociation constant ($K_d$) values of lysocin E for menaquinone and ApoA-I were similar; thus, we assume that a state of equilibrium among them has been established, although it is important to point out that menaquinone is embedded in the membrane and lysocin E binds to lipids. Thus, further detailed analysis is required to elucidate how ApoA-I and ApoA-II impact the pharmacokinetics of lysocin E in the host.

It is important to elucidate the essentiality of lipid II for the ApoA-I-enhancing effect on the antimicrobial activity of lysocin E, but it is difficult to verify using gene disruption strains because lipid II is an essential component for bacterial growth. Competition assays using immobilised menaquinone (Fig. 4a) or lipid II[20] suggested that the binding capacity of lipid II to lysocin E is higher than menaquinone and these compounds could not bind simultaneously. Nevertheless, menaquinone is an essential component for the antimicrobial activity of lysocin E, because the ΔmenA and ΔmenB strains were highly resistant to lysocin E[9] (Fig. 4c). In addition, biochemical analysis using lipid II-containing artificial liposomes suggested that menaquinone, but not lipid II, is required for the membrane disruptive activity of lysocin E (Fig. 4d). Delipidated lipid II did not interact with lysocin E (Fig. 4a), also suggesting that lysocin E could not interact with lipid II embedded in the lipid bilayer. Thus, we assume that menaquinone is required for loading lysocin E into the cell membrane and exerting small membrane disruption activity at sub-MIC. Subsequently, lysocin E transfers to lipid II in the cell membrane, and ApoA-I interacts with the lipid II–lysocin E complex and enhances the membrane damage to kill *S. aureus* (Fig. 4f). This model is also consistent with the findings that lipid II increased lysocin E binding to ApoA-I in the presence of menaquinone (Fig. 2b). Nisin disrupted liposomes containing lipid II as previously reported[26], but not liposomes containing menaquinone (Fig. 4e). Therefore, although the roles of lipid II in the antimicrobial activity of lysocin E and nisin seem to differ, they share a common mechanism by which the antimicrobial activity modulated by ApoA-I via lipid II is enhanced, implying

the existence of a series of similar antimicrobials. Vancomycin[35] and teixobactin[36] are well-known antimicrobials that also bind lipid II. The antimicrobial activity of these antibiotics, however, are not increased by the addition of serum (Supplementary Table 3 and ref. [36]), thus factors other than lipid II binding may be required for ApoA-I-dependent antimicrobial-enhancing activity. Nisin and lysocin E have potent membrane disruptive activity, but vancomycin and teixobactin do not[37], suggesting that the membrane disruption activity might be required for ApoA-I function in antimicrobial binding with lipid II. In addition to ApoA-I, host proteins such as albumin[38], complement[39] and lipids[40] interact with bacteria. Serum proteins are thought to act as inhibitory molecules against the antimicrobial activity of antimicrobial agents. On the other hand, the complement system may increase the susceptibility of *E. coli* to vancomycin[39], and arachidonic acid may enhance the effect of beta-lactam antibiotics and amikacin[40], suggesting that host factors other than ApoA-I contribute to the therapeutic efficacy of antimicrobials. On the basis of our results, we propose that the screening of antimicrobials whose activity is enhanced by host proteins that interact with the bacterial surface is an effective way to discover novel and therapeutically effective antimicrobials with different mechanisms from those of previously identified antimicrobials.

## Methods

**Bacteria, animals, and reagents**. The *S. aureus* RN4220[41], Newman and MSSA1[42] strains were cultured in tryptic soy broth (TSB; Becton Dickinson and Co., Franklin, NJ, USA) overnight at 37 °C with shaking. Various kinds of serum were purchased from MilliporeSigma (St. Louis, MO, USA), heat-inactivated by treatment at 56 °C for 30 min and stored at −20 °C until use. rhApoA-I was purified as described below and the human ApoA-II was purchased from Calbiochem (Merck Millipore, Billerica, MA, USA). Mice (ICR and C57BL/6J) were obtained from and CLEA Japan, Inc. *Apoa1* gene knockout mice were obtained from Jackson Laboratory (Bar Harbor, ME, USA) and bred to the age of 8–12 weeks. Lysocin E was prepared as previously described[9] and lysocin E derivatives were synthesised as previously reported[10]. Lipids, such as PG from egg yolk lecithin, CL from bovine heart and LTA from *S. aureus* were purchased from MilliporeSigma. The other reagents were purchased from Fujifilm Wako Pure Chemicals (Tokyo, Japan) or MilliporeSigma except where noted.

**Measurement of the antimicrobial, bactericidal and protein-binding activities of lysocin E**. The antimicrobial activities of the compounds were measured using the microdilution method[43]. Measurement of the bactericidal activity of lysocin E (0.25 μg ml$^{-1}$) and nisin (2 μg ml$^{-1}$) with or without ApoA-I was performed according to the National Committee for Clinical Laboratory Standards method[44]. We chose the concentration of 25 μg ml$^{-1}$ ApoA-I as the enhancing effect of ApoA-1 on lysocin E was observed in a linear range. For the Δ*menA* and Δ*menB* strains, TSB medium was used instead of cation-adjusted-Mueller Hinton broth (CA-MHB) in the microdilution method. Sera and ApoA-I, II were added to the bacterial culture before distributing them into 96-well plates. ApoA-I and ApoA-II antimicrobial activity were evaluated using the RN4220 strain. The protein binding rate of lysocin E against human plasma was determined by a rapid equilibrium dialysis device (Thermo Fisher Scientific, Waltham, MA, USA) according to the manufacturer's instructions.

**Measurement of membrane-damaging activity**. Membrane-damaging activity of lysocin E and nisin with or without 25 μg ml$^{-1}$ rhApoA-I against *S. aureus* MSSA1 strain was measured using a LIVE/DEAD™ BacLight™ Bacterial Viability kit (Thermo Fisher Scientific) according to the manufacturer's protocol. Membrane potential was measured according to a fluorescence-based assay[9] with slight modification. Briefly, *S. aureus* MSSA1 or Newman strain was grown in TSB at 37 °C with shaking at 200 r.p.m. overnight. The overnight culture was diluted 100-fold in CA-MHB and grown until the $A_{600}$ reached 1.0. The cells were then collected by centrifugation at 7500 × *g* for 10 min at 4 °C, washed with HEPES buffer (pH 7.0) containing 5 mM HEPES, 50 mM glucose and 5 mM EDTA, and resuspended in the same buffer to an $A_{600}$ of 0.05. A membrane potential-sensitive dye, 3,3′-dipropylthiadicarbocyanine iodide, was added to the cells at a final concentration of 250 nM, and the cells were incubated in the dark at 37 °C for 30 min. The fluorescence intensities (excitation 622 nm; emission 670 nm), after adding the respective antimicrobials, were acquired using Spectra Manager ver. 1.55.00 on an FP-6200 spectrofluorometer (JASCO, Tokyo, Japan) equipped with a heated chamber at 37 °C after adding the respective antimicrobials. All experiments were performed under minimal light conditions.

**Purification of lysocin E antimicrobial activity-enhancing factors from calf serum**. Calf serum (10 ml) was added to 15 ml EtOH (60% v/v) and the mixture was centrifuged for 15 min. The collected supernatant was evaporated and suspended in 8 ml Milli-Q water (fraction [fr.] II). Fr. II with formic acid (0.1%) was applied to 10 ml of ODS resin (Waters Corporation, Milford, MA, USA). After washing with 25% EtOH and then 50% EtOH, the resin was eluted with 75% EtOH containing 0.1% formic acid. The eluted sample was evaporated and dissolved in 0.8 ml buffer A (10 mM phosphate buffer [pH 7.1], 6 M urea; Fr. III). Fr. III was applied to the Superose 12 10/300 GL gel filtration column (GE Healthcare) and eluted with the AKTApurifier system (GE Healthcare). The elution was performed in buffer A at a flow rate of 0.5 ml min$^{-1}$ and 60 fractions (0.5 ml per fr.) were collected in 2-ml tubes. All fractions, except for Fr. I, were dialysed in 10 mM phosphate buffer (pH 7.1) before assessment of their activities. Protein concentrations were measured using Coomassie Plus™ protein assay reagent (Thermo Fisher) with bovine serum albumin as the standard. Amino acid sequence analysis was performed by in-gel digestion and mass spectrometric analysis[45]. One unit was defined as the activity that decreases the MIC of lysocin E to 1 μg ml$^{-1}$.

**Plasmid construction, expression and purification of ApoA-I**. The oligonucleotide sequence of the human and mouse apolipoprotein A-I open-reading frames (ORFs), modified for optimised codon usage in *E. coli* (Supplementary Fig. 6) according to a previous report[46], were synthesised at Eurofins Genomics and cloned in the pET20b vector (Novagen). The plasmids, designated pET-hApoWT and pET-mApoWT, were introduced into *E. coli* BL21(DE3)/pLysS and cells were selected with 100 μg ml$^{-1}$ ampicillin and 12.5 μg ml$^{-1}$ chloramphenicol. To construct rhApoA-I variants, we first amplified the two regions within the ORF (nucleotide nos. 1–435 and 481–729) using degenerated primers with linker sequences (Supplementary Table 4) and the pET-hApoWT plasmid as a template. The two products were then connected by another round of PCR and ligated to the *Eco*RV-digested pET20b vector, resulting in pET-hApoΔM. Next, the internal ORF region (nucleotide nos. 196–729) was amplified by PCR using degenerated primers (Supplementary Table 4) and the pET-hApoΔM plasmid as a template. The PCR product was digested with *Nde*I and *Hind*III and then ligated with the pET20b vector linearised with the same restriction enzymes, resulting in pET-hApoΔNM. To construct pET-hApoΔC, the plasmid with C-terminal region-deleted rhApoA-I, we used the primer pair Apo F and ΔC R and pET-hApoWT as a template. Next, we used this template and primers ΔN F and ΔC R to construct pET-hApoΔNC. Colonies were inoculated with NZCYM broth (10 g pancreatic digest casein, 1 g casamino acids, 5 g yeast extract, 5 g NaCl, 1 g magnesium sulfate anhydrous, dissolved in 1 L water and autoclaved at 121 °C for 15 min) in the presence of the above-mentioned antibiotics. After overnight incubation, 0.5 ml of the full-growth cultures were added to 50 ml of the fresh NZCYM broth containing the above-mentioned antibiotics, followed by aerobic incubation at 37 °C to $OD_{600} = 0.6$. Then, 50 μl of 1 M isopropyl β-D-1-thiogalactopyranoside (final concentration 1 mM) was added and the cultures were further incubated for 3 h. Cells were collected by centrifugation, resuspended in 1xGnPB (50 mM phosphate buffer pH 8, 0.5 M NaCl, 3 M guanidine hydrochloride), sonicated and centrifuged at maximum speed for 10 min. The supernatants were collected and applied to Probond, a nickel-chelating resin (Life Technologies). The column was washed successively with 1xGnPB and 20 mM imidazole in 1xGnPB and then eluted with 200 mM imidazole in 1xGnPB. The eluted fractions were dialysed against 50 mM phosphate buffer (pH 8) and 100 mM NaCl. Approximately 5–6 mg of the recombinant protein was obtained from 50 ml of the initial *E. coli* culture.

**Mouse infection experiments**. Mouse protocols followed the Regulations for Animal Care and Use of the University of Tokyo under approval by the Graduate School of Pharmaceutical Science at the University of Tokyo (approval nos. 24–55). Mice were housed in a group of at most five in a cage, kept in a room maintained at 23–24 °C, 55% humidity, and 12 h light/12 h dark cycle. For Table 1, clinically isolated MRSA MR6 strain[47] was cultured overnight, suspended in 7% type II porcine stomach mucin supplemented with 0.2 mM FeNH$_4$-citrate and intraperitoneally administered to mice (ICR, female, 4 weeks of age) at $5.7 \times 10^7$ colony-forming units. Lysocin E (4–0.25 mg kg$^{-1}$), vancomycin (8–0.5 mg kg$^{-1}$), linezolid (8–0.5 mg kg$^{-1}$), or daptomycin (4–0.25 mg kg$^{-1}$) were administered subcutaneously to 5 animals/group at 1 h and 6 h after infection. Mouse survival was assessed after 5 days and ED$_{50}$ values were calculated by multiple logistic regression analysis. For Fig. 1d, the Newman strain was cultured in TSB overnight and cells ($4.0 \times 10^7$ colony-forming units) were washed in phosphate-buffered saline (PBS) were injected into the tail vein of 8–12-week-old female C57BL/6J wild-type mice or *Apoa1* homozygous-deficient mice. After 2 h, lysocin E (0.5 mg kg$^{-1}$) was administered subcutaneously. After 24 h, the kidneys were harvested and crushed with a BioMasher II (Nippi Incorporated, Tokyo, Japan). The suspensions were diluted with PBS(−), plated on TSB agar medium and the number of colonies was counted.

**rhApoA-I pull-down assay**. Menaquinone 4 (MK-4; 100 μl of 1.25 mM) was mixed with N-dodecyl β-D-maltoside (250 μl of 100 mM) in the presence or absence of purified lipid II or lipids (300 μl of 70 μM). Lipid II fraction was prepared from *S. aureus* RN4220 treated with moenomycin, followed by park nucleotide removal[48]. The mixture was dried in vacuo and dissolved in 2 ml of water. The micelles were then filtered

through a 0.45-µm polyvinylidene difluoride membrane filter and a 200 µl aliquot of the filtered micelles were used to evaluate the binding of the antimicrobials in the presence of rhApoA-I. Briefly, 5 µg of antimicrobials was added to the aliquots and the mixture was incubated for 15 min on ice in the presence of 4 µg rhApoA-I. The mixture was then transferred to His-Tag magnetic beads (Dynabeads, Thermo Fisher). The bound protein was then eluted with 250 mM imidazole and then eluted antimicrobials were quantified using an ACQUITY ultra-performance liquid chromatography (UPLC) H-Class PLUS attached to Xevo G2-XS QTof system (Waters Corporation). After injecting the sample into a 2.1 × 100 mm Acquity UPLC HSS T3 1.8 µm column (Waters Corporation) equilibrated with 0.3 mL min⁻¹ of 30% acetonitrile and 0.1% formic acid in water, the column was developed with a linear gradient to 60% acetonitrile and 0.1% formic acid in water over 10 min. The eluate was continuously applied to a Waters Xevo G2-XS QTof mass spectrometer and the spectra were obtained using MassLynx 4.1 (Waters Corporation) in electrospray ionisation (ESI)-positive mode that was analysed by UNIFI Scientific Information System ver. 1.9 (Waters Corporation).

**Lysocin E pull-down assay.** The pull-down assay for the binding of lysocin E and lipid II was performed using Streptavidin beads (Dynabeads™ M-280, Thermo Fisher) and 0.05% [v/v] Tween-20 in PBS. The beads were first washed with the buffer and loaded with premixed 2.5 µg biotinylated lysocin in 30 µl of 50 µM lipid II. After mixing and incubating at room temperature for 15 min, the beads were washed twice with the buffer and eluted with 6 M guanidine thiocyanate. Control experiments, without lysocin E, were performed to control for the possibility of nonspecific binding. The amount of lipid II in the eluted sample was quantified using UPLC-MS.

**Bio-layer interferometry analysis.** Bio-layer interferometry (BLI) measurements were performed on a BLItz system (ForteBio, Fremont, CA, USA) maintained at room temperature. Kinetic analyses were performed using BLItz Pro ver. 1.2.1.5 (ForteBio). The biotinylated ligand, diluted to 0.1 mg ml⁻¹ in BLItz buffer (0.05% [v/v] Tween-20 in PBS), was immobilised on a streptavidin biosensor and the kinetic analysis was performed by running BLItz buffer for 300 s each, in the presence and absence of analyte for association and dissociation, respectively. Biotinylated menaquinone (**5**) and biotinylated lysocin E derivatives (**3** and **4**) were synthesised as described in the Supplementary Information (Supplementary Figs. 7–15).

**Amount of rhApoA-I binding on the surface of S. aureus.** *Staphylococcus aureus* RN4220 strain was grown overnight in TSB at 37 °C with shaking at 150 r.p.m. The $A_{600}$ was adjusted to 0.5 with CA-MHB, 1 ml of which was treated with 0, 10, 20, or 40 µg ml⁻¹ of rhApoA-I and further incubated at 37 °C with shaking at 150 r.p.m. for 5 min. The cells were then centrifuged at 900 × g at 4 °C for 30 min, washed twice with PBS, and then the pellets were suspended in SDS-PAGE sample buffer. Under these conditions, the *S. aureus* cells were not lysed. The supernatants were collected, boiled, and applied to 12.5% SDS-PAGE. The gel was stained with Coomassie brilliant blue, and the band intensities were quantified using ImageJ software (1.47v, NIH, USA)[49].

**Liposome disruption assay.** We prepared liposomes by making some modifications to the previously described procedure[9]. To 30 µl PG (50 mM, dissolved in methanol/chloroform) contained in a 10 mL round-bottom flask, 7.5 µl of CL (50 mM, dissolved in the same solvent) and 500 µL of chloroform were added. To this mixture, lipids (1.0 mol % MK-4 and/or 0.1 mol % lipid II) were added to prepare liposomes. Control liposomes without lipids were also prepared using the same protocol. A lipid film prepared was rehydrated with 5 mM sodium HEPES (pH 7.5) containing 2.5 mM calcein. The liposome suspension was freeze-thawed five times and filtered through a Sephadex G-50 column with 20 mM sodium HEPES (pH 7.5) containing 1 mM EDTA to remove the free calcein. The obtained liposomes contained 40 ± 10% lipid II. The liposome fraction was diluted 5-fold with buffer, and the increase in fluorescence intensity (excitation: 310 nm; emission: 530 nm) was measured by a spectrofluorometer after adding the antimicrobials. The fluorescence value for 100% disruption was estimated by adding Triton X-100 to each sample.

**Quantification of lipid II.** The lipid II fraction was prepared from 5 ml cultures as previously described[20] and hydrolysed by incubating at 99 °C for 30 min in 100 µl of 10 mM ammonium acetate (pH 4.2). The samples were freeze-dried and dissolved in 30 µl water. The supernatant, after centrifugation at 16,000 × g for 10 min, was analysed by a quantitative liquid chromatography-mass spectrometry. The liquid chromatography-mass spectrometry system comprised an ACQUITY UPLC H-Class PLUS and Xevo G2-XS QTof system and CORTECS UPLC T3 1.6 µm 2.1 × 150 mm columns (Waters Corporation) maintained at 40 °C with a flow rate of 0.3 ml min⁻¹. Solvent A was 0.1% aqueous formic acid and solvent B was 0.1% formic acid in acetonitrile. After 8 µl of the sample was applied to the column equilibrated with solvent A, the column was developed with the same solvent for 2 min, followed by a linear gradient of 100–80% of A over 15 min. Mass spectra were collected in ESI-negative mode using MassLynx 4.1 (Waters Corporation) and analysed by UNIFI Scientific Information System (ver. 1.9). The peak area of delipidated lipid II (Supplementary Fig. 1d; **7**) to assess the exact mass *m/z* 664.255

$[M-2H]^{2-}$ (calculated mass *m/z* 664.256 $[M-2H]^{2-}$) was quantified and the structure was confirmed by the fragment pattern generated from MS$^E$ analysis using the UNIFI Scientific Information System. To evaluate the lipid II purity of the fraction, we purified lipid II referring to a previous report[50]. Briefly, a lipid II fraction prepared as above was dissolved in 75% methanol with 25% 50 mM ammonium bicarbonate and separated using an Alliance HPLC system with a PDA detector 2998 and Empower 3 software (Waters Corporation) and PEGASIL C4 SP100 column (4.6 × 250 mm, Senshu Scientific, Tokyo, Japan). Solvent A was 50 mM ammonium bicarbonate and solvent B was 100% methanol. After injecting the lipid II fraction prepared as above, the column was developed with a linear gradient of 85–90% of B over 15 min, followed by 90–100% of B over 5 min, and isocratic 100% B for 5 min. The lipid II peak appeared at 11.3 min and was collected. The purity of this fraction was analysed by developing 70–100% of A over 30 min and isocratic 100% B for 10 min and found to be >95% (Supplementary Fig. 16a), and the structure was confirmed by in-fusion MS/MS (Supplementary Fig. 16b, c).

**Statistical analysis.** Statistical analysis was performed using GraphPad Prism 8.4.3 (GraphPad Software, San Diego, CA, USA). *P* values of 0.05 or less were considered significant.

**Reporting summary.** Further information on research design is available in the Nature Research Reporting Summary linked to this article.

## Data availability
The data that support the findings of this study are available from the corresponding author upon reasonable request. Source data are provided with this paper.

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

## Acknowledgements

This work was supported by JSPS KAKENHI Grant Numbers 19K07140JP, 26102714 and 24689008, the Mochida Memorial Foundation for Medical and Pharmaceutical Research and the Takeda Science Foundation (to H.H.), and in part by JSPS KAKENHI Grant Numbers 20K16253, 21H02733 and 15H05783 and Drug Discovery Support Promotion Project from AMED: Japan Agency for Medical Research and Development to K.S., A.M. and H.H., and JP17F17421, TBRF and IFO fellowships to S.P. and K.S. We thank the Genome Pharmaceutical Institute Co., Ltd for preparing the lysocin E and silkworms.

## Author contributions

H.H. conceived the idea. H.H and S.P. designed the experiments. S.P. analysed the interactions between ApoA-I, lipid II, and lysocin E. A.P., J.S., K.I., and J.Y. purified the ApoA-I and assayed the antimicrobial activity. H.H., S.P. and A.P. performed the membrane disruption assay. H.I. and K.T. synthesised the lysocin E derivatives and performed the structural analysis of lipid II. M.I. and K.S. provided critical discussion for writing the manuscript. A.M. purified lipid II and performed the liposome disruption assay. H.H., S.P., and A.P. wrote the manuscript. All authors contributed critically to the interpretation of the results and gave final approval for publication.

## Competing interests

K.S. is a consultant for Genome Pharmaceutical Institute Co., Ltd. The remaining authors declare no competing interests.
