## [Peer Review File · Nature Communications]

Serum apolipoprotein A-I potentiates the therapeutic efficacy of lysocin E against *Staphylococcus aureus*REVIEWER COMMENTS

Reviewer #1 (Remarks to the Author):

This is an exciting, elegant and novel study on the role of apoA-I in strongly enhancing the effect of the new antimicrobial agent lysocin E. The antimicrobial activity of apolipoproteins has been known for some time, but often in terms of binding to membranes, either in bacteria or their isolated components. While binding to and disruption of membranes is robust, direct activity has been modest, and the antimicrobial activity of apolipoproteins is poorly understood. Through an elaborate panel of experimentation, the authors were able to convincingly show that serum proteins apoA-I, and to a lesser extent apoA-II, are potent contributors to the antimicrobial activity against *S. aureus*. They proposed that apoA-I is magnifying the effect of Lysocin E interaction with lipid II, causing major disruptions of the bacterial membrane integrity, leading to cell death. This is a very plausible model, and is reminiscent of a detergent like activity of apoA-I. ApoA-I is very effective to bind to packing defects on lipoprotein surfaces, membranes, and bilayer vesicles. The membrane disruption by solubilization as a mechanism for their antimicrobial activity has been proposed for insect apolipophorin III, however, detailed experimental evidence in vivo has been sparse. Thus, this manuscript is an important step forward understanding the role serum proteins, and in particular apolipoprotein A-I, play in innate immunity.

One concern is that the authors constructed a truncation variant, named apoA-I Δ NM. According to the authors, the lipid binding domain was removed. First, most of apoA-I is able to bind to lipids, so the terminology of lipid binding domain is ambiguous. Second, it would help if the authors clearly describe with amino acids have been removed in this truncation variant (in the result section). Based on the description in the Methods, it seems that the first 65 residues and residues 146 to 160 were removed from apoA-I (I could be off by a couple of amino acids). What was the rationale for removing these specific residues? The C-terminal domain (residues ~ 190 to 243) is considered the part of the protein that initiates lipid binding, aided by its relatively unstructured nature and the adoption of α -helical structure upon lipid binding. Then other α -helices follow suit and make direct contact with lipids.

A related point is that the C-terminal part of the protein has been identified as an important site for antimicrobial activity against another gram-negative bacterium using a deletion variant Δ 220-243 (Apolipoprotein A-I exerts bactericidal activity against *Yersinia enterocolitica* Serotype O:3*, 2011, *J. Biol. Chem* 268, 38211). Similar to the present study, the antimicrobial activity of apoA-I was not direct but needed the complement system. The discussion and wider scope of their findings will be strengthened if the authors include a brief discussion on this.

Minor points

Since the major focus of the study is apoA-I, the authors may consider including this into the title.

Line 86, purified human apoA-II was used, it is not clear if this also recombinant protein or isolated from plasma (and how it was isolated).

In the Discussion, line 258, it is mentioned that lipophorin has similar functions as apoA-I. However, the lipophorin studied was apolipophorin I and II, which are apoB like apolipoproteins in insects with masses of 220 and 74 kDa, and not to be confused with the much smaller apolipophorin III (~18 kDa). Apolipophorin III shares similar structural and functional characteristics compared with apoA-I.

Figure 3 a. It is not clear to what apoA-I is compared to in the Lysocin E and Nisin treatments (Viecle?).

Reviewer #2 (Remarks to the Author):

The manuscript „Serum proteins potentiate therapeutic effect of lysocin E against *S. aureus* “ by Hamamoto et al. reports on the impact of Apolipoprotein A1 to increase the antimicrobial activity of the antibiotic lysocin E, previously reported to interact with menaquinone and/or lipid II. The authors claim to provide first evidence that antimicrobial activity can be potentiated through interactions of host derived factors, i.e. serum protein ApoA1, and microbial components, i.e. lipid II. At first glance the study appears interesting. However, the data presented do not support the proposed model. While the manuscript is well written and the (individual) results are properly described the model derived is superficial and flawed. Importantly the authors appear to ignore current knowledge on the multifactorial roles of apolipoproteins. Although the main role of HDL and its principal apolipoproteins has long been considered to be its participation in reverse cholesterol transport and its anti-atherogenic effect, more recent studies have involved these proteins in other defensive functions in mammals, such as antiviral, antimicrobial and anti-inflammatory activities. Apolipoproteins are crucial factors of the innate defense and share characteristic features with cationic antimicrobial peptides and their abilities to bind to bacterial membranes and to adopt specific secondary structures in membrane environments, an essential prerequisite to their attachment and insertion into bacterial membranes.

The role of menaquinone is still uncertain and it is unclear if lipid II binding to ApoA-I, given it is specific, is the reason for the potentiating effect. As shown for e.g. albumin, the binding capacity is strongly influenced by fatty acid and other factors. Therefore, other lipids such as phosphatidylglycerol might have the same effect and are basically co-factors for ApoA-I in binding lysocin-E. The interaction of apolipoproteins with (phospho)lipids has been demonstrated (e.g. lipoteichoic acid, sphingomyelin, LPS, cardiolipin, phosphatidylglycerol).

I deem it necessary to provide further experimental data on the overall binding capacity of ApoA-I for lysocin-E, lipid II and most importantly other lipids mentioned from the authors themselves. Otherwise the model is highly speculative and not sufficiently supported by the experimental data. In terms of consistency, the authors should also stick with one species as protein binding is species dependent. Therefore, it would have been better to use heterologously produced mouse ApoA-I and -II for the in vitro experiments. The authors use human recombinant ApoA-I and -II (fig. 1b), isolation was done from *Bos taurus* (ext. data fig. 2) and in vivo experiments were done in mice. It is very difficult to pool these results and extract the correct information.

Specific comments:

Page 2, 37-41:

The authors claim to demonstrate the potentiating effect of serum components on antimicrobial activity for the first time. The potentiating effect of serum on antimicrobial activity was already shown for 3. generation cephalosporins in 1988. <https://pubmed.ncbi.nlm.nih.gov/2496656/> Although they aim to connect novelty to an interaction with an additional microbial component, namely lipid II, does not justify this statement. Firstly, the authors could not provide conclusive data on a specific interaction with lipid II. Second, a direct correlation between the observed therapeutic effect and the proposed players involved (i.e. ApoA1, Lysocin E, lipid II) is not proven.

Page 2, 46-48:

The authors claim that protein binding (PB) is generally considered a problem in terms of antimicrobial efficacy.

The effect of PB on compound activity is controversially discussed and still matter of ongoing research. Regarding antibiotics, it was shown that Daptomycin has a PB of over 90%, but its clinical efficacy is not impaired as predicted from the in vitro data, because they are administered at higher doses than necessary. The same is true for telavancin where a high PB has no effect on clinical efficacy.

Furthermore, PB can also act as a depot for certain substances, which in the case for antibiotics, can keep the concentration above the MIC for a longer period of time.

Authors should take care in using words such as “generally” if there is ongoing research on the topic.

Page 2, 53-59, Table 1:

I am unsure if comparing in vitro MIC with ED50 values is appropriate, if there is no information on PB of lysocin E provided. Furthermore, it would be necessary to include MIC values using MHB in the presence of serum or albumin. As shown for telavancin, the MIC for a MRSA strain increases

10fold in the presence of serum or albumin in contrast to MHB alone.

<https://www.ncbi.nlm.nih.gov/pmc/articles/PMC478526/>

It is also difficult to compare ED50 values if there is no information on pharmacokinetics of lysocin E. Elimination processes might be significantly different for each drug.

Extended Data Fig 1 c:

It is not clear why the structure of delipidated lipid II is shown. This was used to determine the concentration of natural lipid II isolated from staphylococcal cells. I recommend to show full length lipid II.

Page 5: Extended Data Figure 1 (referenced in lines 79,84, 86, 102) should read Extended Data Fig 2.

Figure 1a:

The impact of serum on the antimicrobial activity of daptomycin and vancomycin was tested. While Ext Data Table 2 includes nisin and vancomycin, daptomycin was not included. Daptomycin should also be tested in presence of ApoA1. What is the rationale for choosing 25 µg/ml ApoA1? Why did the authors use human plasma instead of human serum when comparing the effects on antimicrobial activity? While there is no change in MIC comparing plasma with serum values are questionable.

Figure 1b,c:

The panel shows that ApoA-I and ApoA-II both potentiate the effect of lysocin E in vitro, yet in vivo the effect was attributed to ApoA-I. While this might be a concentration dependent effect (7fold less ApoA-II), I would also include ApoA2 gene knock-out mice to rule out ApoA-II is necessary for the ApoA-I effect on lysocin E activity.

Do the authors have any explanation for the lack of in vivo activity of ApoA-II?

S. aureus Newman was used in the mouse model. Include MIC in Ext Data Table 1.

Recommend to include a control where the authors show the effect of serum in the absence of lysocin E

Ext. Figure 2:

There are additional bands visible between 36 and 64 kDa. It is recommended to identify these proteins as well.

Page 10, line 141:

It is not surprising that ApoA1 binds to *S. aureus*, as ApoA1 interacts with membranes and was shown to interact with LTA. Only relative numbers are given. How much ApoA1 did bind to cells? The authors should further provide raw data.

Ext. Data Fig 4:

The authors used BLI for determination of binding parameters. Please provide evidence that the BLITZ system is suitable for the determination of binding parameters of small molecules, such as lipid II. Nisin could be used as a control and binding parameters have been determined by different methods.

Binding parameters of lysocin E to lipid II indicate no high affinity interaction, compared to other lipid II binding antibiotics, e.g. Nisin Kd 2.68×10^{-7} M (DOPC + Lipid II) ; Nisin Kd 1.03×10^{-6} M (DOPC only).

Ext. Data Fig 4d: How was the pull down assay performed? Controls should be included.

Ext. Data Fig. 5:

In lines 266-268 the authors correctly mention, that ApoA-I binds to phosphatidylglycerol (PG) and cardiolipin (CL), therefore, I do not understand why they did not use vesicles made of PG and included CL. This would be an easy experiment and would provide data on the necessity and specificity of lipid II.

Page 17, 278 ff:

The authors fail to provide evidence for the role of menaquinone in their model. As shown by

Santiago et al. the binding sites of MK and lipid II overlap and the high level of resistance towards lysocine E in menA and menB mutants is attributed to the slow growth of these strains. A slow growth also means that less lipid II is present and therefore, the effect of lysocin E is diminished. <https://www.ncbi.nlm.nih.gov/pmc/articles/PMC5964011/pdf/nihms946863.pdf>
Such effects have further been reported for *S. aureus* small colony variants.

It is more likely that the lipid tail is responsible for ApoA-I intercalation. While both, MK and lipid II, contain a lipid tail, the one of lipid II might bind more strongly to ApoA-I making MK the less favorable interaction partner.

Fig. 3b: Typo "viecle"

Ext. Data Fig. 2: The figure is not referenced in the text (see comment above)

Figure 3:

3a: Values are given as fold change/increment. It would be important to show how much nisin bound. This should be correlated to the amount of lipid II and vancomycin.

For the latter it would be also crucial to show which lipid II species (including pentaglycine and amidation ?) was used? Vancomycin shows different binding affinities to modified vs unmodified lipid II. The authors should further include controls, e.g. cardiolipin or phosphatidylglycerol.

Vancomycin binds to the D-ala-ala terminus of lipid II and does not insert into the membrane (compared to nisin). Vancomycin is not the ideal compound for comparison.

Lipid II: Please provide evidence on the identity and purity of lipid II by mass spectrometry. It is further recommended to use more direct methods to determine the concentration of lipid II. The authors incorporated lipid II into liposomes. It should be determined how much lipid II has been incorporated.

Figure 3d: Why was ApoA1 not included in the membrane potential measurements, in presence and absence of antibiotics?

Page 11, 185 ff: These methodologies cannot be compared! Any conclusion is obsolete.

Reviewer #3 (Remarks to the Author):

This manuscript by Hamamoto et al provides a mechanistic understanding of how the in vivo efficacy of the novel Lysocin E antibiotic is greater than would be expected on the basis of its in vitro activity towards methicillin resistant *Staphylococcus aureus* (MRSA). Although interaction with host proteins commonly reduces the in vivo efficacy of antibiotics, the minimum effective dose of Lysocin E, a lipopeptide antibiotic, was potentiated through its interaction with the serum protein ApoA-1. The in vivo relevance was further demonstrated using ApoA-1 deficient transgenic mice, in which the enhanced in vivo efficacy of Lysocin E was no longer evident. The authors further reveal that the interaction of lysocin E with ApoA-1 was enhanced by Lipid II, a lipid carrier of peptidoglycan subunits that is essential for cell wall synthesis. This adds to previous knowledge that lysocin E targets menaquinone, which is an essential component of the electron transport chain in Gram positive bacteria. The authors present a model whereby interaction of lysocin E with ApoA-1 in serum promotes greater interaction with both menaquinone (MK) and lipid II, causing maximum membrane disruption. The work is carefully done with a strong combination of advanced biochemistry techniques and infection models. There is support for the authors claim of being the first to demonstrate that antimicrobial activity can be potentiated through interactions of host serum proteins with microbial components to enhance the therapeutic effect. However, some of the claims seem to be contradictory, or stated in a manner that the meaning is unclear.

Comments and questions for the authors to consider:

1. The present work identifies lipid II as a target for Lysocin E, and the abstract claims that the binding capacity of Lysocin E to ApoA-1 was enhanced by lipid II. This is somewhat confusing,

since lines 142 to 145 state that the enhancing effect of ApoA-1 cannot be explained by increased accumulation of lysocin E to the cell surface, while lines 171-172 state that "lipid II increased the binding of lysocin E to ApoA-1 in a pull down assay in the presence of menaquinone". This seems to suggest that ApoA-1 can promote increased accumulation of lysocin E. Taking this into account, could the authors please provide a temporal view of how lysocin E is bactericidal during antimicrobial therapy? For example, ApoA-1 is in serum, and lysocin E has a binding constant of 3.6 μM for ApoA-1 compared to 4 μM for menaquinone (MK). How would this relate to therapeutic levels of lysocin E in blood, and should we assume that during antimicrobial therapy, lysocin E forms a complex with ApoA-1 before it encounters a microbial surface? Once this complex engages lipid II on a microbial surface, could it not then recruit more lysocin E?

2. Would it be possible for the authors to image lysocin E on the microbial surface in the presence and absence of ApoA-1? For example, they have been successful in biotinylating lysocin E. Presumably a fluorescent streptavidin derivative could be used to quantify lysocin E binding to cell surfaces when bacteria are pre-treated with ApoA-1 compared to non-treated cells. Assessing overall fluorescent intensity, accompanied by microscopy to visualize fluorescence localization might provide some valuable mechanistic detail.

3. ApoA-1 has a lipid binding domain, and its ability to enhance the MIC of lysocin E is attenuated when this lipid binding domain is deleted. It is presumed that this is due to loss of binding to lipid II. However, lysocin E is also a lipopeptide. The authors could consider some assays to assess the binding constant of lysocin E for the ΔNM variant of ApoA-1, and also assaying for the ability of this variant to bind to *S. aureus* cells. Is it strongly reduced?

4. The authors use Nisin as an example of another antimicrobial that interacts with lipid II and report that as with lysocin E, the activity of Nisin is also enhanced by ApoA-1. Teixobactin is a newly described antimicrobial that also targets lipid II. Could the authors comment on whether ApoA-1 would also be expected to potentiate the activity of Teixobactin? The discovery of Teixobactin was also published in Nature, and it is a significant omission that this is not mentioned.

Minor comments:

5. Binding of lysocin E to ApoA-1 was conducted in the presence of menaquinone. Should the authors have also tried menaquinone as a co-factor in binding of lysocin E to lipid II?

6. Extended data Table 1 shows MIC data for menA and menB deficient strains of *S. aureus*, but there is no mention of this in the text.

Reviewer #4 (Remarks to the Author):

The manuscript by Hamamoto represents a very interesting and original report showing that the activity of a recently identified antibiotic (lysocin E) is strongly potentiated by a host protein. Overall, the data are of strong significance to the field of infectious disease/microbiologists and could have an impact on the development of antibiotics.

Overall the paper is well-written, although the paper would benefit from a more thorough introduction and better-balanced discussion also discussing other reports suggesting synergy between host immune components and antibiotics.

Main comments:

1. Number of strains used to show synergy between ApoA-I and lysocin E is limited.

In Extended Data Table 1, only four *S. aureus* strains were tested (including an ATCC strain and a laboratory strain (RN4220)). This makes it difficult to understand how broadly applicable these data are for clinically relevant *S. aureus* strains. Did the authors just pick *S. aureus* strains for which this works? It would be more convincing if the authors also include clinical isolates, including both MSSA and MRSA (including the highly virulent USA300).

Suggest to determine whether synergy between a) ApoA-I and lysocin E AND b) bovine serum and

lysocin E also occurs for these other *S. aureus* strains?

2. The authors give the impression that they are the first to show that components in serum can enhance the activity of antibiotics. However, this is wrong and there should be a more balanced discussion also stating other literature that showed such synergy. Some examples of papers that should be referred to:

*Pruul et al. Potentiation of antibacterial activity of azithromycin and other macrolides by normal human serum. *Antimicrob Agents Chemother* . 1992 Jan;36(1):10-6.

*Heesterbeek et al. Complement-dependent outer membrane perturbation sensitizes Gram-negative bacteria to Gram-positive specific antibiotics. *Sci Rep*. 2019 Feb 28;9(1):3074.

*Giamarellos-Bourboulis et al, Ex vivo synergy of arachidonate-enriched serum with ceftazidime and amikacin on multidrug-resistant *Pseudomonas aeruginosa*. *J Antimicrob Chemother*. 2003 Feb;51(2):423-6

3. The introduction is really short. The paper would benefit from a more balanced introduction, for instance including info about the antibiotic lysocin E and *S. aureus*. Also, it is not clear from the introduction whether the data in Table 1 are already published in their earlier work or are they new? In that case it should be clearer that they are part of the Results section.

Minor comments:

4. Page 5, line 90: "the addition of purified bovine and human ApoA-I or II did not inhibit the growth of *S. aureus* at up to 300 $\mu\text{g ml}^{-1}$."

I would like to see these data included but it is not clear if they are added or not?

5. In the discussion the authors claim: 'Proteins in serum are thought to act principally as inhibitory molecules against the antimicrobial activity of antimicrobial agents.'. This statement should be modified and other studies showing synergy between serum proteins and antibiotics should not be ignored. See comment 2.

Response to reviewers

We thank the reviewers for their critical comments, which have helped us to greatly improve our manuscript. The figure and table legends in the revised manuscript were modified to conform to the *Nature Communications* format.

Reviewer #1 (Remarks to the Author):

This is an exciting, elegant and novel study on the role of apoA-I in strongly enhancing the effect of the new antimicrobial agent lysocin E. The antimicrobial activity of apolipoproteins has been known for some time, but often in terms of binding to membranes, either in bacteria or their isolated components. While binding to and disruption of membranes is robust, direct activity has been modest, and the antimicrobial activity of apolipoproteins is poorly understood. Through an elaborate panel of experimentation, the authors were able to convincingly show that serum proteins apoA-I, and to a lesser extent apoA-II, are potent contributors to the antimicrobial activity against *S. aureus*. They proposed that apoA-I is magnifying the effect of Lysocin E interaction with lipid II, causing major disruptions of the bacterial membrane integrity, leading to cell death. This is a very plausible model, and is reminiscent of a detergent like activity of apoA-I. ApoA-I is very effective to bind to packing defects on lipoprotein surfaces, membranes, and bilayer vesicles. The membrane disruption by solubilization as a mechanism for their antimicrobial activity has been proposed for insect apolipophorin III, however, detailed experimental evidence in vivo has been sparse. Thus, this manuscript is an important step forward understanding the role serum proteins, and in particular apolipoprotein A-I, play in innate immunity.

Thank you for your positive comments. We improved the manuscript according to your comments.

One concern is that the authors constructed a truncation variant, named apoA-I Δ NM. According to the authors, the lipid binding domain was removed. First, most of apoA-I is able to bind to lipids, so the terminology of lipid binding domain is ambiguous. Second, it would help if the authors clearly describe with amino acids have been removed in this truncation variant (in the result section). Based on the description in the Methods, it seems that the first 65 residues and residues 146 to 160 were removed from apoA-I (I could be off by a couple of amino acids). What was the rationale for removing these specific residues? The C-terminal domain (residues ~ 190 to 243) is considered the part of the protein that initiates lipid binding, aided by its relatively unstructured nature and the adoption of α -helical structure upon lipid binding. Then other α -helices follow suit and make direct contact with lipids.

We designed a truncated variant according to references 25, 26, and 27 (listed below). We actually compared several deletion mutants and found that the Δ NM mutant had the smallest enhancing effect on the antimicrobial activity of lysocin E. These data are now included as Supplementary Figure 5b. In addition, according to your suggestion, we changed the term “lipid binding domain” to a more suitable term (line 201, and legend of Fig. 2e) and specified the deleted amino acids in the Results section (line 201-202 and Supplementary Fig. 5a).

25. Frank, P. G. & Marcel, Y. L. Apolipoprotein A-I: structure-function relationships. *J Lipid Res* 41, 853-872 (2000).

26. Tanaka, M. et al. Contributions of the N- and C-terminal helical segments to the lipid-free structure and lipid interaction of apolipoprotein A-I. *Biochemistry* 45, 10351-10358 (2006).

27. Biedzka-Sarek, M. et al. Apolipoprotein A-I exerts bactericidal activity against *Yersinia enterocolitica* serotype O:3. *J Biol Chem* 286, 38211-38219 (2011).

A related point is that the C-terminal part of the protein has been identified as an important site for antimicrobial activity against another gram-negative bacterium using a deletion variant Δ 220-243 (Apolipoprotein A-I exerts bactericidal activity against *Yersinia enterocolitica* Serotype O:3*, 2011, *J. Biol. Chem* 268, 38211). Similar to the present study, the antimicrobial activity of apoA-I was not

direct but needed the complement system. The discussion and wider scope of their findings will be strengthened if the authors include a brief discussion on this.

We think ApoA-I directly enhances lysocin E activity for the following reasons: 1) adding ApoA-I to the medium alone decreased the MIC of lysocin E without the addition of serum, and 2) ApoA-I bound to the surface of *S. aureus*. 3) serum we used in this study is heat inactivated. Furthermore, deletion of the C-terminal region reduces the enhancing effect of ApoA-I on the antimicrobial activity of lysocin E. Related discussion was added in line 357-369.

Minor points

Since the major focus of the study is apoA-I, the authors may consider including this into the title.

We revised the title of the revised manuscript to include Apolipoprotein A-I.

Line 86, purified human apoA-II was used, it is not clear if this also recombinant protein or isolated from plasma (and how it was isolated).

Purified human ApoA-II isolated from human plasma was purchased from Calbiochem as written in the Methods section. We revised the sentence to improve the clarity (now line 94-95 and 540).

In the Discussion, line 258, it is mentioned that lipophorin has similar functions as apoA-I. However, the lipophorin studied was apolipophorin I and II, which are apoB like apolipoproteins in insects with masses of 220 and 74 kDa, and not to be confused with the much smaller apolipophorin III (~18 kDa). Apolipophorin III shares similar structural and functional characteristics compared with apoA-I.

Thank you for pointing this out. These sentences were modified in the revised manuscript (now line 345-348).

Figure 3 a. It is not clear to what apoA-I is compared to in the Lysocin E and Nisin treatments (Viecle?).

Thank you for this comment. We meant “Vehicle”, which is N-dodecyl β -D-maltoside without lipid II or lipids. In response to Reviewer #2, we modified Figure 3a and b. For Figure 3b (now Fig. 3c in revised manuscript), we performed this membrane damaging assay using fluorescent dye by dose response manner instead of the classical method of measuring the UV absorbance of the genomic DNA leak, and thus the figure has been modified in the revised manuscript.

Reviewer #2 (Remarks to the Author):

The manuscript „Serum proteins potentiate therapeutic effect of lysocin E against *S. aureus*“ by Hamamoto et al. reports on the impact of Apolipoprotein A1 to increase the antimicrobial activity of the antibiotic lysocin E, previously reported to interact with menaquinone and/or lipid II. The authors claim to provide first evidence that antimicrobial activity can be potentiated through interactions of host derived factors, i.e. serum protein ApoA1, and microbial components, i.e. lipid II. At first glance the study appears interesting. However, the data presented do not support the proposed model. While the manuscript is well written and the (individual) results are properly described the model derived is superficial and flawed. Importantly the authors appear to ignore current knowledge on the multifactorial roles of apolipoproteins. Although the main role of HDL and its principal apolipoproteins has long been considered to be its participation in reverse cholesterol transport and its anti-atherogenic effect, more recent studies have involved these proteins in other defensive functions in mammals, such as antiviral, antimicrobial and anti-inflammatory activities. Apolipoproteins are crucial factors of the innate defense and share characteristic features with cationic antimicrobial peptides and their abilities to bind to bacterial membranes and to adopt specific secondary structures

in membrane environments, an essential prerequisite to their attachment and insertion into bacterial membranes.

The role of menaquinone is still uncertain and it is unclear if lipid II binding to ApoA-I, given it is specific, is the reason for the potentiating effect. As shown for e.g. albumin, the binding capacity is strongly influenced by fatty acid and other factors. Therefore, other lipids such as phosphatidylglycerol might have the same effect and are basically co-factors for ApoA-I in binding lysocin-E. The interaction of apolipoproteins with (phospho)lipids has been demonstrated (e.g. lipoteichoic acid, sphingomyelin, LPS, cardiolipin, phosphatidylglycerol). I deem it necessary to provide further experimental data on the overall binding capacity of ApoA-I for lysocin-E, lipid II and most importantly other lipids mentioned from the authors themselves. Otherwise the model is highly speculative and not sufficiently supported by the experimental data.

Thank you for providing this important viewpoint. According to your suggestions, we performed additional experiments and added the results to the revised manuscript.

In terms of consistency, the authors should also stick with one species as protein binding is species dependent. Therefore, it would have been better to use heterologously produced mouse ApoA-I and – II for the in vitro experiments. The authors use human recombinant ApoA-I and -II (fig. 1b), isolation was done from *Bos taurus* (ext. data fig. 2) and in vivo experiments were done in mice. It is very difficult to pool these results and extract the correct information.

We understand that we need to use different species of ApoA-I depending on the aim of the analysis. Bovine serum is often used for antimicrobial activity test. Since purification of proteins required large amount and bovine serum is readily available than other sera, it is often used for identifying responsible proteins. For genetics studies analyzing the therapeutic application of lysocin E in mammals, however, mice must be used. Further, when assessing clinical applications, it is important to evaluate human recombinant proteins. To address this issue, we compared the activities of different ApoA-I used in this study to show the similarities in function using prepared recombinant mouse ApoA-I protein (rmApoA-I). We found that rmApoA-I reduced the MIC of lysocin E against *S. aureus* in a similar manner as recombinant human ApoA-I (Fig. 1b). We also found that ApoA-I and ApoA-II exhibited similar specific activities among various species (Supplementary Fig. 2a, e). We concluded that bovine, mouse, and human ApoA-I and ApoA-II have principally the same functions, and thus it is possible to discuss the lysocin E antimicrobial-enhancing activity across species. These results are discussed in line 96-97 and 341-345.

Specific comments:

Page 2, 37-41:

The authors claim to demonstrate the potentiating effect of serum components on antimicrobial activity for the first time. The potentiating effect of serum on antimicrobial activity was already shown for 3. generation cephalosporins in 1988. <https://pubmed.ncbi.nlm.nih.gov/2496656/> Although they aim to connect novelty to an interaction with an additional microbial component, namely lipid II, does not justify this statement. Firstly, the authors could not provide conclusive data on a specific interaction with lipid II. Second, a direct correlation between the observed therapeutic effect and the proposed players involved (i.e. ApoA1, Lysocin E, lipid II) is not proven.

We agree with these comments and revised the sentence as follows; “The antimicrobial activity of lysocin E is potentiated through interactions of host serum proteins with microbial components. Understanding these mechanisms broadens the available strategies for developing antimicrobials by taking advantage of host-microbe interactions.”

Page 2, 46-48:

The authors claim that protein binding (PB) is generally considered a problem in terms of

antimicrobial efficacy.

The effect of PB on compound activity is controversially discussed and still matter of ongoing research. Regarding antibiotics, it was shown that Daptomycin has a PB of over 90%, but its clinical efficacy is not impaired as predicted from the in vitro data, because they are administered at higher doses than necessary. The same is true for telavancin where a high PB has no effect on clinical efficacy.

Furthermore, PB can also act as a depot for certain substances, which in the case for antibiotics, can keep the concentration above the MIC for a longer period of time.

Authors should take care in using words such as “generally” if there is ongoing research on the topic.

We thank you for your comments. We revised our manuscript according to the suggestions (line 41-49) and removed the term “generally”.

Page 2, 53-59, Table 1:

I am unsure if comparing in vitro MIC with ED50 values is appropriate, if there is no information on PB of lysocin E provided. Furthermore, it would be necessary to include MIC values using MHB in the presence of serum or albumin. As shown for telavancin, the MIC for a MRSA strain increases 10fold in the presence of serum or albumin in contrast to MHB alone.

<https://www.ncbi.nlm.nih.gov/pmc/articles/PMC478526/>

According to your suggestion, we added data regarding the protein binding capacity of lysocin E using human plasma and MIC data in 10% bovine serum in Table 1. The results suggested that lysocin E has high protein binding capacity, but the addition of sera reduced the MIC against *S. aureus* by more than 60-fold (Fig. 1). A related sentence in the text (line 77-79) was modified.

It also difficult to compare ED50 values if there is no information on pharmacokinetics of lysocin E. Elimination processes might be significantly different for each drug.

[Redacted]

Extended Data Fig 1 c:

It is not clear why the structure of delipidated lipid II is shown. This was used to determine the concentration of natural lipid II isolated from staphylococcal cells. I recommend to show full length lipid II.

According to your suggestions, we now provide full-length lipid II as Extended Data Fig 1c (now Supplementary Fig. 1c).

Page 5: Extended Data Figure 1 (referenced in lines 79,84, 86, 102) should read Extended Data Fig 2.

Thank you for pointing this out. The figure number was corrected and is now formatted correctly for *Nature Communications*.

Figure 1a:

The impact of serum on the antimicrobial activity of daptomycin and vancomycin was tested. While Ext Data Table 2 includes nisin and vancomycin, daptomycin was not included. Daptomycin should also be tested in presence of ApoA1. What is the rationale for choosing 25 µg/ml ApoA1? Why did the authors use human plasma instead of human serum when comparing the effects on antimicrobial activity? While there is no change in MIC comparing plasma with serum values are questionable.

We added the MIC data of daptomycin in the presence of ApoA-I in Supplementary Table 3. We selected a 25-µg ml⁻¹ concentration of ApoA-I as the enhancing effect of ApoA-I on lysocin E was observed in a linear range. We added a comment on this in the revised manuscript (line 553-554).

In going through our records, we noticed that the term “human plasma” was an incorrect description. We actually purchased “human serum” from MilliporeSigma (cat no. H4522) and used it for the experiment; we never used human plasma in any of the studies reported in this manuscript. Thus, we modified Figure 1 and related sentences (line 29, 80-81, 131). Thank you for giving opportunity to correct our error.

Figure 1b,c:

The panel shows that ApoA-I and ApoA-II both potentiate the effect of lysocin E in vitro, yet in vivo the effect was attributed to ApoA-I. While this might be a concentration dependent effect (7fold less ApoA-II), I would also include ApoA2 gene knock-out mice to rule out ApoA-II is necessary for the ApoA-I effect on lysocin E activity.

Do the authors have any explanation for the lack of in vivo activity of ApoA-II?

S. aureus Newman was used in the mouse model. Include MIC in Ext Data Table 1.

Recommend to include a control where the authors show the effect of serum in the absence of lysocin E

We understand your point regarding the ApoA-II knockout mice. Unfortunately, it is very difficult for us to perform experiments using *apoA2* gene knockout mice, because our present facility is not approved to handle genetically modified animals. To perform the necessary tests of ApoA-II for ApoA-I function, we further tested the combined effect of ApoA-I and ApoA-II against the MIC of lysocin E. The results demonstrated no synergistic effect between them, suggesting that these factors work in an independent manner (Supplementary Fig. 2g). Thus, we concluded that the remaining activity of the serum in *apoA1* gene knockout mice could be explained by residual ApoA-II, and we think ApoA-II activity is only a minor part of the serum activity. As pointed out, we did not elucidate the function of ApoA-II in this study. We mention this point in the Results and Discussion section (line 111-114, 375-378). We added the combined effects of ApoA-I and ApoA-II as Supplementary Fig. 2g and, modified the related sentence to emphasize the ApoA-II activity in the *apoA1* knockout mice in line 117-118.

As suggested, we added the MIC data for the Newman strain in Supplementary Table 1, and the effect of serum against *S. aureus* in Supplementary Figure 2f.

Ext. Figure 2:

There are additional bands visible between 36 and 64 kDa. It is recommended to identify these proteins as well.

These peaks were not correlated with the elution pattern of the activity in the gel filtration chromatography, suggesting these proteins were not corresponding to lysocin E enhancing activity in the serum. Thus, we do not think identification of these proteins is necessary at the present stage.

Page 10, line 141:

It is not surprising that ApoA1 binds to *S. aureus*, as ApoA1 interacts with membranes and was shown to interact with LTA. Only relative numbers are given. How much ApoA1 did bind to cells? The authors should further provide raw data.

We modified Figure 2a and displayed representative SDS-PAGE image were also added in Supplementary Figure 3a according with your suggestion. We modified the sentences (line 164) to emphasis our intent.

Ext. Data Fig 4:

The authors used BLI for determination of binding parameters. Please provide evidence that the BLITZ system is suitable for the determination of binding parameters of small molecules, such as lipid II. Nisin could be used as a control and binding parameters have been determined by different methods. Binding parameters of lysocin E to lipid II indicate no high affinity interaction, compared to other lipid II binding antibiotics, e.g. Nisin Kd 2.68×10^{-7} M (DOPC + Lipid II) ; Nisin Kd 1.03×10^{-6} M (DOPC only).

[Redacted]

Ext. Data Fig 4d: How was the pull down assay performed? Controls should be included.

The method is now described in the Methods section of the main text (lines 706-714). An experiment without lysocin E was performed as a control.

Ext. Data Fig. 5:

In lines 266-268 the authors correctly mention that ApoA-I binds to phosphatidylglycerol (PG) and cardiolipin (CL), therefore, I do not understand why they did not use vesicles made of PG and included CL. This would be an easy experiment and would provide data on the necessity and specificity of lipid II.

According to this suggestion, we performed the same experiments using lipid vesicles comprising PG and CL, and obtained results consistent with those using DOPC vesicles. These results suggest that menaquinone, not lipid II, is a critical factor for the membrane disruption induced by lysocin E. In the case of nisin, membrane disruption was achieved by lipid II, not by menaquinone. The revised data were moved to Figure 4d,e.

Page 17, 278 ff:

The authors fail to provide evidence for the role of menaquinone in their model. As shown by Santiago *et al.* the binding sites of MK and lipid II overlap and the high level of resistance towards lysocin E in *menA* and *menB* mutants is attributed to the slow growth of these strains. A slow growth also means that less lipid II is present and therefore, the effect of lysocin E is diminished.

<https://www.ncbi.nlm.nih.gov/pmc/articles/PMC5964011/pdf/nihms946863.pdf>

Such effects have further been reported for *S. aureus* small colony variants.

Thank you for your critical comments. We performed a competition assay using biotinylated menaquinone. The results suggested that lysocin E bound more strongly to lipid II than menaquinone, consistent with the results of Santiago *et al.* Therefore, menaquinone and lipid II compete for binding with lysocin E (Figure 4a in revised manuscript). The lipid II content was reduced in $\Delta menA$ and $\Delta menB$, but the amount required for the nisin antimicrobial activity was retained (Fig. 4b, c), while the antimicrobial activity of lysocin E against the $\Delta menA$ mutant was completely lost. In addition, externally added lipid II did not disrupt the antimicrobial activity of lysocin E, but the antimicrobial activity of nisin was decreased (Supplementary Table 3). Furthermore, addition of lipid II attenuated the enhancing activity of ApoA-I against the antimicrobial activity of lysocin E (Supplementary Table 3). A liposome disruption assay also suggested that menaquinone is a binding partner of lysocin E with respect to membrane disruption. In addition, delipidated lipid II could not elute lysocin E-bound menaquinone (Fig. 4a). From these results, we assume that lysocin E cannot interact with lipid II at the outer membrane side. Thus, we hypothesized that menaquinone is required for loading lysocin E in/on the membrane and the initial membrane disruption.

We also inserted a new subheading (line 295-319) and Figure 4 regarding role of the menaquinone, added MIC data in Supplementary Table 3, modified the Discussion section (line 387-400), and describe the presumed model more clearly (now moved to Fig. 4f).

It is more likely that the lipid tail is responsible for ApoA-I intercalation. While both, MK and lipid II, contain a lipid tail, the one of lipid II might bind more strongly to ApoA-I making MK the less favorable interaction partner.

We agree that lipid II is a favorable interaction partner with ApoA-I for lysocin E. The above results suggested that the lipid tail of lipid II is important for lysocin E interaction and menaquinone is not a binding partner for the ApoA-I interaction. In addition, we found that lipid II attenuated ApoA-I

activity against the MIC reduction of lysocin E (Supplementary Table 3), suggesting that lipid II interacted with ApoA-I. We included this data and discussed the finding in line 222-225 and 227.

Fig. 3b: Typo “viecle”

Thank you for your comment, though, we changed this figure according to your following another comment.

Ext. Data Fig. 2: The figure is not referenced in the text (see comment above)

Thank you for your comment. The figure (now Supplementary Fig. 2) is now correctly referred in the revised manuscript.

Figure 3:

3a: Values are given as fold change/increment. It would be important to show how much nisin bound. This should be correlated to the amount of lipid II and vancomycin.

For the latter it would be also crucial to show which lipid II species (including pentaglycine and amidation ?) was used? Vancomycin shows different binding affinities to modified vs unmodified lipid II. The authors should further include controls, e.g. cardiolipin or phosphatidylglycerol.

Vancomycin binds to the D-ala-ala terminus of lipid II and does not insert into the membrane (compared to nisin). Vancomycin is not the ideal compound for comparison.

According to the suggestions, we changed the units of the Y-axis and include other controls such as PG, CL, and LTA of lipid II, and removed the vancomycin data in Figure 3a. Related sentences were modified (line 213). In response to your first general comments, we also performed the same experiments with other controls, such as PG, CL, and LTA, for lysocin E and modified Fig. 2c. We used lipid II without amidation and included pentaglycine as shown in Supplementary Figure 1. We confirmed the structure by infusion MS/MS analysis as shown in Supplementary Figure 7b,c.

Lipid II: Please provide evidence on the identity and purity of lipid II by mass spectrometry. It is further recommended to use more direct methods to determine the concentration of lipid II.

The authors incorporated lipid II into liposomes. It should be determined how much lipid II has been incorporated.

Thank you for your critical comments. According to your comment, we established a direct detection method for natural lipid II according to *J Am Chem Soc* 124, 3656-3660, doi:10.1021/ja017386d (2002) using a C4 column (Materials and Methods section line 769-780). This method is not applicable to LC-MS, however, as we could not find a UPLC-column that allowed us to elute full-length lipid II and the detection sensitivity in MS differs depending on the compound structure. Thus, we performed HPLC to test the purity of lipid II, and confirmed the lipid II structure by in-fusion mass spectrometry (MS/MS) as in Supplementary Figure 7b, c. Furthermore, we determined the incorporated lipid II content in liposomes using this system as described in the Methods section (line 746).

Figure 3d: Why was ApoA-I not included in the membrane potential measurements, in presence and absence of antibiotics? Page 11, 185 ff: These methodologies cannot be compared! Any conclusion is obsolete.

We performed the experiment by adding ApoA-I as suggested by the reviewer (Fig. 3d). Furthermore, we performed the membrane damaging assay again using a LIVE/DEAD™ BacLight™ Bacterial Viability Kit instead of DNA leakage comparable to the membrane potential loss assay. We found that membrane damaging activity and loss of membrane potential against *S. aureus* were induced partially at sub-MIC levels of both antimicrobials and increased by the addition of ApoA-I. These results support the notion that ApoA-I promotes the membrane-disrupting activity of lysocin E and nisin at

sub-MIC levels. We modified Figure 3b, d (now Fig. 3c, d, Fig 3c changed to Fig. 3b) and related sentences (line 215-222).

Reviewer #3 (Remarks to the Author):

This manuscript by Hamamoto et al provides a mechanistic understanding of how the in vivo efficacy of the novel Lysocin E antibiotic is greater than would be expected on the basis of its in vitro activity towards methicillin resistant *Staphylococcus aureus* (MRSA). Although interaction with host proteins commonly reduces the in vivo efficacy of antibiotics, the minimum effective dose of Lysocin E, a lipopeptide antibiotic, was potentiated through its interaction with the serum protein ApoA-1. The in vivo relevance was further demonstrated using ApoA-1 deficient transgenic mice, in which the enhanced in vivo efficacy of Lysocin E was no longer evident. The authors further reveal that the interaction of lysocin E with ApoA-1 was enhanced by Lipid II, a lipid carrier of peptidoglycan subunits that is essential for cell wall synthesis. This adds to previous knowledge that lysocin E targets menaquinone, which is an essential component of the electron transport chain in Gram positive bacteria. The authors present a model whereby interaction of lysocin E with ApoA-1 in serum promotes greater interaction with both menaquinone (MK) and lipid II, causing maximum membrane disruption. The work is carefully done with a strong combination of advanced biochemistry techniques and infection models. There is support for the authors claim of being the first to demonstrate that antimicrobial activity can be potentiated through interactions of host serum proteins with microbial components to enhance the therapeutic effect. However, some of the claims seem to be contradictory, or stated in a manner that the meaning is unclear.

Thank you for your comprehensive summary and understanding of our research. Our replies to your comments are provided below.

Comments and questions for the authors to consider:

1. The present work identifies lipid II as a target for Lysocin E, and the abstract claims that the binding capacity of Lysocin E to ApoA-1 was enhanced by lipid II. This is somewhat confusing, since lines 142 to 145 state that the enhancing effect of ApoA-1 cannot be explained by increased accumulation of lysocin E to the cell surface, while lines 171-172 state that “lipid II increased the binding of lysocin E to ApoA-1 in a pull down assay in the presence of menaquinone”. This seems to suggest that ApoA-1 can promote increased accumulation of lysocin E. Taking this into account, could the authors please provide a temporal view of how lysocin E is bactericidal during antimicrobial therapy? For example, ApoA-1 is in serum, and lysocin E has a binding constant of 3.6 μM for ApoA-1 compared to 4 μM for menaquinone (MK). How would this relate to therapeutic levels of lysocin E in blood, and should we assume that during antimicrobial therapy, lysocin E forms a complex with ApoA-1 before it encounters a microbial surface?

Thank you for your critical advice. We agree that statements “ApoA-I did not accumulate lysocin E on the cell surface of *S. aureus*” and “binding of lysocin E to ApoA-I enhanced by lipid II in presence of menaquinone” are contradictory. We performed new experiments and showed that ApoA-I did not decrease the required minimal concentration of lysocin E for the membrane damaging assay (Fig. 3cd), which also suggests that ApoA-I does not act to increase the local concentration of lysocin E. Based on these results and the mechanism presented in Figure 4f, lysocin E loaded in the membrane depending on the presence of menaquinone. Then, lysocin E is transferred to lipid II in the inner membrane, and ApoA-I interacts with the lysocin E and lipid II complex. These observations do not rule out the possibility that lysocin E forms a complex with ApoA-I before it encounters a microbial surface, but ApoA-I comprises HDL in the blood, and further studies are needed to investigate the actual effects of HDL against lysocin E. Furthermore, the K_d values of lysocin E for MK and ApoA-I were in the same range; thus, we assume that a state of equilibrium among them was established. Further, we need to point out that the interaction of lysocin E and menaquinone in an actual situation should be investigated, as menaquinone is embedded in the membrane. Related sentences were added to the Discussion section (line 371-383).

2. Would it be possible for the authors to image lysocin E on the microbial surface in the presence and absence of ApoA-1? For example, they have been successful in biotinylating lysocin E. Presumably a fluorescent streptavidin derivative could be used to quantify lysocin E binding to cell surfaces when bacteria are pre-treated with ApoA-1 compared to non-treated cells. Assessing overall fluorescent intensity, accompanied by microscopy to visualize fluorescence localization might provide some valuable mechanistic detail.

[Redacted]

3. ApoA-1 has a lipid binding domain, and its ability to enhance the MIC of lysocin E is attenuated when this lipid binding domain is deleted. It is presumed that this is due to loss of binding to lipid II. However, lysocin E is also a lipopeptide. The authors could consider some assays to assess the binding constant of lysocin E for the Δ NM variant of ApoA-1, and also assaying for the ability of this variant to bind to *S. aureus* cells. Is it strongly reduced?

According to your suggestion, we performed a kinetic analysis of rhApoA-I Δ NM to lysocin E and found that the K_d values of these interactions were in the same range (Supplementary Fig. 4a). Binding of rhApoA-I Δ NM to the cell surface of *S. aureus* was significantly (not strongly) reduced compared with wild-type (Fig. 2a). These findings imply that these domains are involved in the interactions of ApoA-I with the bacterial surface and contribute to reduce the antimicrobial enhancing activity of lysocin E. We added these data to Figure 2a and Supplementary Figure 4a, and modified the related sentences (line 203-205).

4. The authors use Nisin as an example of another antimicrobial that interacts with lipid II and report that as with lysocin E, the activity of Nisin is also enhanced by ApoA-1. Teixobactin is a newly described antimicrobial that also targets lipid II. Could the authors comment on whether ApoA-1 would also be expected to potentiate the activity of Teixobactin? The discovery of Teixobactin was also published in Nature, and it is a significant omission that this is not mentioned.

The antimicrobial activity of teixobactin is not changed by the addition of serum. Therefore, we assume that the antimicrobial activity of teixobactin is not increased by the addition of ApoA-I. We mentioned teixobactin and discussed the difference between teixobactin and lysocin E in the Discussion section, line 406-412.

Minor comments:

5. Binding of lysocin E to ApoA-1 was conducted in the presence of menaquinone. Should the authors have also tried menaquinone as a co-factor in binding of lysocin E to lipid II?

We used a competition assay to demonstrate that lysocin E does not simultaneously bind to lipid II and menaquinone (Fig.4a). Thus, we assume that menaquinone is not a co-factor in the binding of lysocin E to lipid II. Discussion regarding this point was added to the revised version of the manuscript (line 387-401).

6. Extended data Table 1 shows MIC data for menA and menB deficient strains of *S. aureus*, but there is no mention of this in the text.

Thank you for your careful reading of our manuscript. We mentioned the MIC data for *menA*- and *menB*-deficient strains of *S. aureus* in the revised manuscript (line 390-391, now moved to Fig. 4c).

Reviewer #4 (Remarks to the Author):

The manuscript by Hamamoto represents a very interesting and original report showing that the activity of a recently identified antibiotic (lysocin E) is strongly potentiated by a host protein. Overall, the data are of strong significance to the field of infectious disease/microbiologists and could have an impact on the development of antibiotics.

Overall the paper is well-written, although the paper would benefit from a more thorough introduction and better-balanced discussion also discussing other reports suggesting synergy between host immune components and antibiotics.

Thank you for your positive comments. We modified the Introduction and Discussion sections according to your advice. Our responses to your comments are shown below.

Main comments:

1. Number of strains used to show synergy between ApoA-I and lysocin E is limited.

In Extended Data Table 1, only four *S. aureus* strains were tested (including an ATCC strain and a laboratory strain (RN4220)). This makes it difficult to understand how broadly applicable these data are for clinically relevant *S. aureus* strains. Did the authors just pick *S. aureus* strains for which this works? It would be more convincing if the authors also include clinical isolates, including both MSSA and MRSA (including the highly virulent USA300).

Suggest to determine whether synergy between a) ApoA-I and lysocin E AND b) bovine serum and lysocin E also occurs for these other *S. aureus* strains?

We examined effect of ApoA-I and bovine serum against various MSSA and MRSA strains including USA300 JE2 strain (Supplementary Table 1 and 2), and added results in line 79-80 and 99-100.

2. The authors give the impression that they are the first to show that components in serum can enhance the activity of antibiotics. However, this is wrong and there should be a more balanced discussion also stating other literature that showed such synergy. Some examples of papers that should be referred to:

*Pruul et al. Potentiation of antibacterial activity of azithromycin and other macrolides by normal human serum. *Antimicrob Agents Chemother* . 1992 Jan;36(1):10-6.

*Heesterbeek et al. Complement-dependent outer membrane perturbation sensitizes Gram-negative

bacteria to Gram-positive specific antibiotics. Sci Rep. 2019 Feb 28;9(1):3074.

*Giamarellos-Bourboulis et al, Ex vivo synergy of arachidonate-enriched serum with ceftazidime and amikacin on multidrug-resistant *Pseudomonas aeruginosa*. J Antimicrob Chemother. 2003 Feb;51(2):423-6

Thank you for your comments. We referred to previous studies reporting antibiotics whose activities were enhanced by host factors in the Introduction and Discussion sections, and included the references you suggested (line 41-49 and 415-418).

3. The introduction is really short. The paper would benefit from a more balanced introduction, for instance including info about the antibiotic lysocin E and *S. aureus*. Also, it is not clear from the introduction whether the data in Table 1 are already published in their earlier work or are they new? In that case it should be clearer that they are part of the Results section.

According to your suggestion, we amended the Introduction section (line 54-63). Table 1 also shows new data using MRSA strain. We moved the table to the Results section (line 74-77).

Minor comments:

4. Page 5, line 90: “the addition of purified bovine and human ApoA-I or II did not inhibit the growth of *S. aureus* at up to 300 $\mu\text{g ml}^{-1}$.”

I would like to see these data included but it is not clear if they are added or not?

We now present the antimicrobial activity data using rhApoA-I and hApoA-II in Supplementary Figure 2f.

5. In the discussion the authors claim: ‘Proteins in serum are thought to act principally as inhibitory molecules against the antimicrobial activity of antimicrobial agents.’

This statement should be modified and other studies showing synergy between serum proteins and antibiotics should not be ignored. See comment 2.

Thank you for your comment. We modified the Discussion section in relation to comment 2 in line 414-418.

REVIEWER COMMENTS

Reviewer #1 (Remarks to the Author):

The authors have appropriately addressed the concerns and revised the manuscript. The authors now included data of the activity of several other deletion variants of rhApoA-I, which helps to understand why the NM deletion mutant was used and had the lowest activity. I still have concerns about the description of this mutant.

On page 5, line 201, the authors describe that the domains for lipid binding have been removed even though the rebuttal states that alternative wording has been used. There are two problems with this statement. First, from a protein structure perspective, these are not domains, but could be defined as regions or helices. Take in mind that apoA-I is considered by many as a two-domain protein, residues 1 – 190 form the N-terminal helix bundle domain, while residues 190 to 243 form a relatively unstructured C-terminal domain. For a recent description of the complex structure of apoA-I please I would like to refer to

“J.T. Melchior, R.G. Walker, A.L. Cooke, J. Morris, M. Castleberry, T.B. Thompson, M.K. Jones, H.D. Song, K.-A. Rye, M.N. Oda, M.G. Sorci-Thomas, M.J. Thomas, J.W. Heinecke, X. Mei, D. Atkinson, J.P. Segrest, S. Lund-Katz, M.C. Phillips, W.S. Davidson, A Consensus Model of Human Apolipoprotein A-I in its Monomeric and Lipid-free State, *Nat Struct Mol Biol*, 24 (2017) 1093–1099.”

Further, the regions deleted in the NM mutant are not necessarily the lipid binding region. While most of the protein can be engaged in lipid binding, residues 44-65 and 222–243 possess the highest lipid binding activity. Further, residues 143-164 are involved in LCAT activation, and this section has been deleted in the NM mutant. Therefore, the description that the lipid binding domain(region) has been removed is too simplistic. By taking out the N-terminal 65 residues and internal residues 146-160, the structure and thus function of the protein is likely severely affected. The review by Mei and Atkinson (*Archives of Medical Research* 46 (2015) 351-360), has a clear description of the various regions in apoA-I and their functional role, and the authors should use that information.

Minor point, Supplemental Figure 2 legend, it would help the reader if panel C describes that fraction 17-21 contains apoA-I and fractions 24 contain apoA-II.

Reviewer #2 (Remarks to the Author):

I have gone through the revised manuscript and appreciate the efforts made by the authors to address the remaining questions to the best of their ability and feasibility. The authors have addressed most of the questions sufficiently by performing additional experiments and providing appropriate controls and explanations.

However, I would like the authors to address one remaining concern: In the previous assessment I deemed it necessary to provide further experimental data on the overall binding capacity of ApoAI for additional lipids. To address this the authors performed pull-down assays (Figure 2c). In contrast to the experiment presented in Figure 2b, this experiment was performed in the absence of MK. However, as stated in line 194 ff “We further demonstrated that lipid II increased the binding of the natural-type lysocin E to rhApoA-I (Fig. 2b) in a pull-down assay in the presence of menaquinone, and this phenomenon was not observed in the absence of menaquinone (Fig. 2c). In addition, other ApoA-I binding lipids, such as phosphatidylglycerol23, cardiolipin24, and lipoteichoic acid 21, which are present in the *S. aureus* membrane did not increase natural-type lysocin E binding to rhApoA-I (Fig. 2c).” increased lipid II binding capacity was only observed in the presence of MK. That said, it would be important to analyze the effect of other lipids in the presence of MK and compare to the effect in absence of MK. Otherwise data provided in Figures 2b

and 2c are difficult to correlate.

In addition, the observed increment changes (Fig 2c) are not significant except for cardiolipin. Should it read $p= 0.0495$?

Please state the source of lipids in the material and method section.

Reviewer #3 (Remarks to the Author):

This manuscript describes lysocin E as a class of antimicrobial agent for which in vivo activity is potentiated through interaction with host proteins. The authors provide compelling evidence that ApoA1 interacts with lysocin E and microbial lipid II to cause disruption of the microbial cell membrane. This appears to be a complex interaction that is facilitated through initial loading of lysocin E into the membrane, transfer of lysocin E to lipid II, and binding of ApoA1 to this lysocin E-lipid II complex to cause massive membrane disruption. The authors data are largely supportive of this model, though the explanations can be somewhat confusing.

The authors have comprehensively addressed comments from a previous review and the manuscript is greatly improved. The authors should consider writing a full description of the model that is presented in Fig. 4f. This is critical to understanding the paper, but there is no description of the model in the Legend to Fig. 4. There are also some minor spelling errors in labels for Fig 4f (Laoding, Menaqunone > Loading, Menaquinone)

Reviewer #4 (Remarks to the Author):

The authors have appropriately addressed all the comments of this reviewer. This paper is of high significance to the field and the methodologies are excellent.

I only have one minor question to rephrase a sentence in the introduction.

In the Introduction the authors state:

"For example, some host factors, such as serum albumin or complement, bind to some antimicrobials, which influences their activity."

To the best of my knowledge complement factors do not 'bind directly to antimicrobials'. Instead they form pores in bacterial membranes which allow antibiotics to enter the cell (ref 40)

Response to reviewers

We thank all reviewers for their further evaluation of our manuscript and their suggestions to improve it. We modified the manuscript in accordance with the following comments.

REVIEWER COMMENTS

Reviewer #1 (Remarks to the Author):

The authors have appropriately addressed the concerns and revised the manuscript. The authors now included data of the activity of several other deletion variants of rhApoA-I, which helps to understand why the NM deletion mutant was used and had the lowest activity. I still have concerns about the description of this mutant.

On page 5, line 201, the authors describe that the domains for lipid binding have been removed even though the rebuttal states that alternative wording has been used. There are two problems with this statement. First, from a protein structure perspective, these are not domains, but could be defined as regions or helices. Take in mind that apoA-I is considered by many as a two-domain protein, residues 1 – 190 form the N-terminal helix bundle domain, while residues 190 to 243 form a relatively unstructured C-terminal domain. For a recent description of the complex structure of apoA-I please I would like to refer to

“J.T. Melchior, R.G. Walker, A.L. Cooke, J. Morris, M. Castleberry, T.B. Thompson, M.K. Jones, H.D. Song, K.-A. Rye, M.N. Oda, M.G. Sorci-Thomas, M.J. Thomas, J.W. Heinecke, X. Mei, D. Atkinson, J.P. Segrest, S. Lund-Katz, M.C. Phillips, W.S. Davidson, A Consensus Model of Human Apolipoprotein A-I in its Monomeric and Lipid-free State, *Nat Struct Mol Biol*, 24 (2017) 1093–1099.”

Further, the regions deleted in the NM mutant are not necessarily the lipid binding region. While most of the protein can be engaged in lipid binding, residues 44-65 and 222—243 possess the highest lipid binding activity. Further, residues 143-164 are involved in LCAT activation, and this section has been deleted in the NM mutant. Therefore, the description that the lipid binding domain(region) has been removed is too simplistic. By taking out the N-terminal 65 residues and internal residues 146-160, the structure and thus function of the protein is likely severely affected. The review by Mei and Atkinson (*Archives of Medical Research* 46 (2015) 351-360), has a clear description of the various regions in apoA-I and their functional role, and the authors should use that information.

We deeply appreciate the critical discussion regarding the structural issues of ApoA-I. We replaced “domain” with “region” in the revised manuscript. We carefully read the suggested manuscripts and modified related sentences in the Results (lines 205-211) and Discussion (lines 382-393).

Minor point, Supplemental Figure 2 legend, it would help the reader if panel C describes that fraction 17-21 contains apoA-I and fractions 24 contain apoA-II.

Thank you. We denoted the presence of ApoA-I and ApoA-II in panel c.

Reviewer #2 (Remarks to the Author):

I have gone through the revised manuscript and appreciate the efforts made by the authors to address the remaining questions to the best of their ability and feasibility. The authors have addressed most of the questions sufficiently by performing additional experiments and providing appropriate controls and explanations.

Thank you for your important comments. The manuscript was improved dramatically by your critical advice.

However, I would like the authors to address one remaining concern: In the previous assessment I deemed it necessary to provide further experimental data on the overall binding capacity of ApoAI for additional lipids. To address this the authors performed pull-down assays (Figure 2c). In contrast to the experiment presented in Figure 2b, this experiment was performed in the absence of MK. However, as stated in line 194 ff “We further demonstrated that lipid II increased the binding of the natural-type lysocin E to rhApoA-I (Fig. 2b) in a pull-down assay in the presence of menaquinone, and this phenomenon was not observed in the absence of menaquinone (Fig. 2c). In addition, other ApoA-I binding lipids, such as phosphatidylglycerol²³, cardiolipin²⁴, and lipoteichoic acid²¹, which are present in the *S. aureus* membrane did not increase natural-type lysocin E binding to rhApoA-I (Fig. 2c).” increased lipid II binding capacity was only observed in the presence of MK. That said, it would be important to analyze the effect of other lipids in the presence of MK and compare to the effect in absence of MK. Otherwise data provided in Figures 2b and 2c are difficult to correlate.

We agree with this comment. According to this comment, we performed the binding assay in the presence of MK for all other lipids tested. The results suggested that lipid II specifically enhanced the binding of lysocin E to ApoA-I in the presence of menaquinone. We modified Fig. 2b and related sentences (line 203-204).

In addition, the observed increment changes (Fig 2c) are not significant except for cardiolipin.

Should it read $p=0.0495$?

Yes, we modified the figure.

Please state the source of lipids in the material and method section.

According to your comments, we added the source of lipid (line 564-566).

Reviewer #3 (Remarks to the Author):

This manuscript describes lysocin E as a class of antimicrobial agent for which in vivo activity is potentiated through interaction with host proteins. The authors provide compelling evidence that ApoA1 interacts with lysocin E and microbial lipid II to cause disruption of the microbial cell membrane. This appears to be a complex interaction that is facilitated through initial loading of lysocin E into the membrane, transfer of lysocin E to lipid II, and binding of ApoA1 to this lysocin E-lipid II complex to cause massive membrane disruption. The authors data are largely supportive of this model, though the explanations can be somewhat confusing.

The authors have comprehensively addressed comments from a previous review and the manuscript is greatly improved. The authors should consider writing a full description of the model that is presented in Fig. 4f. This is critical to understanding the paper, but there is no description of the model in the Legend to Fig. 4. There are also some minor spelling errors in labels for Fig 4f (Laoding, Menaqunone > Loading, Menaquinone)

Thank you for your insightful comments, which greatly helped us to improve our manuscript. According to your suggestion, we added an explanation in the legend for Fig. 4f. In addition, we modified the typographical errors in Fig. 4f.

Reviewer #4 (Remarks to the Author):

The authors have appropriately addressed all the comments of this reviewer. This paper is of high significance to the field and the methodologies are excellent.

I only have one minor question to rephrase a sentence in the introduction.

In the Introduction the authors state:

“For example, some host factors, such as serum albumin or complement, bind to some antimicrobials, which influences their activity.”

To the best of my knowledge complement factors do not ‘bind directly to antimicrobials’. Instead they form pores in bacterial membranes which allow antibiotics to enter the cell (ref 40)

Thank you for this comment. We deleted the word “complement” from this sentence.